# Learning Causally Invariant Reward Functions from Diverse Demonstrations

## Abstract

Inverse reinforcement learning methods aim to retrieve the reward function of a Markov decision process based on a dataset of expert demonstrations. The commonplace scarcity and heterogeneous sources of such demonstrations can lead to the absorption of spurious correlations in the data by the learned reward function. Consequently, this adaptation often exhibits behavioural overfitting to the expert data set when a policy is trained on the obtained reward function under distribution shift of the environment dynamics. In this work, we explore a novel regularization approach for inverse reinforcement learning methods based on the *causal invariance* principle with the goal of improved reward function generalization. By applying this regularization to both exact and approximate formulations of the learning task, we demonstrate superior policy performance when trained using the recovered reward functions in a transfer setting [1].

## 1 Introduction

In the domain of reinforcement learning, the formulation of a suitable reward function plays a pivotal role in shaping the behaviour of decision making agents. This is commonly justified by the widely adopted belief that the reward function is a succinct representation of a task goal in a given environment specified as a Markov decision process (MDP) (Ng et al., 2000). Eliciting the correct behavioural policies via the optimization of a reward function is of paramount importance for the deployment of RL agents to real world domains such as various robotics scenarios (Pomerleau, 1991; Billard et al., 2008) or expert behaviour forecasting (Kitani et al., 2012). However, the challenge of designing such a function typically entails a cumbersome and error-prone process of handcrafting a heuristic reward signal that accounts for all the intricacies of the task at hand.

Inverse reinforcement learning (IRL) methods aim to solve the problem of inferring the reward function of an MDP based on a dataset of temporal behaviours. These trajectories are typically obtained from an agent that is assumed to demonstrate near-optimal performance in the respective MDP. There are multiple benefits to learning an explicit reward function compared to alternatives such as behavioural cloning (Pomerleau, 1991) or other imitation approaches (Ho & Ermon, 2016) including the ability to transfer the reward function across problems and enhanced robustness to compounding errors (Swamy et al., 2021).

The IRL problem is challenging due to a number of factors. The problem of recovering the reward from the statistics of expert trajectories is generally ill-posed, as there typically exist many reward functions which satisfy the optimization constraints (Ng et al., 1999). While this property is effectively tackled by regularization in the form of a maximum entropy objective (Ziebart et al., 2008; 2010), scaling up IRL to handle large-scale problems remains a challenge. In particular, it requires a variational formulation dependent on non-linear function approximation methods (Finn et al., 2016a; Fu et al., 2017). Since the amount of available expert data is typically limited, this can lead to overfitting phenomena, which are particularly pronounced in highly parameterized models such as neural networks (Ying, 2019; Song et al., 2019). An additional difficulty arises when there is a significant discrepancy between expert demonstrations originating from different experts. We show that pooling expert demonstrations in one dataset under the same label introduces spurious correlations which are absorbed in the representation of the reward function. By

---

[1]The implementation is provided at: **\<removed for anonymity\>**

optimizing the expected cumulative reward defined by such functions, the agent might fail to learn meaningful behaviours due to optimization of the spurious correlations present in the reward model.

In order to circumvent this issue, we consider causal properties of the reward learning problem. The causal invariance principle (Peters et al., 2015; Heinze-Deml et al., 2017; Arjovsky et al., 2019) studies the generalization problem of supervised learning models through the lens of causality. It postulates that the conditional distribution of the target variable must be asymptotically stable across samples obtained under observational and interventional settings of the data generating process. By only considering causally invariant representations of the input, this ensures avoiding the reliance on spurious correlations in the model predictions. We propose to adapt this principle for the context of reward function learning, where we aim to elicit behavioural policies without exploiting spurious reward features, i.e. features which would prevent the reward from providing a meaningful training signal under distribution shift. To achieve this goal, we make the assumption that variations in expert demonstrations are a product of causal interventions on the data generating process of trajectories. Under this assumption, the conditional distribution describing the optimality of a transition must be stable for experts demonstrations gathered on a specific task. By applying the causal invariance principle under this assumption, we show that we can recover reward functions which are invariant across a population of experts and demonstrate improve generalized w.r.t. certain types of distribution shift.

**Contributions.** Our contributions are as follows: (i) we formulate the assumption that the variations between experts performing considered to perform near-optimally on the same task can be seen as interventional settings of the underlying trajectory distribution and (ii) propose a regularization principle for inverse reinforcement learning methods based on the principle of causal invariance. This modelling choice allows us to learn reward functions that are invariant to spurious correlations between the transitions and optimality label present in the expert data. We (iii) demonstrate the efficacy of this approach in both tractable, finite state-spaces, which we refer to as the exact setting as well as large continuous state-spaces, which we denote as the approximate setting. In the first setting, we visually analyze the recovered reward functions and verify their invariance properties w.r.t. the input data. In the second setting, we demonstrate improved ground truth performance when a policy is trained using the regularized reward in MDPs with perturbed dynamics.

## 2 Method

In this section, we describe our method. Section 2.1 presents the problem setting of learning rewards in the maximum entropy IRL setting (Ziebart et al., 2008) in both primal and dual form and reviews the connection to a class of adversarial optimization methods based on distribution matching. In section 2.2, we outline how spurious correlations arise in the context of the IRL problem and connect this to the causal invariance principle. Section 2.3 shows how to incorporate this principle as a regularization strategy for reward learning.

### 2.1 Problem setting

We begin by giving an overview of the problem setting. We consider environments modelled by a *Markov decision process* $(\mathcal{S}, \mathcal{A}, \mathcal{T}, \mu_0, R, \gamma)$, where $\mathcal{S}$ is the state space, $\mathcal{A}$ is the action space, $\mathcal{T}$ is the family of transition distributions on $\mathcal{S}$ indexed by $\mathcal{S} \times \mathcal{A}$ with $p(s'|s, a)$ describing the probability of transitioning to state $s'$ when taking action $a$ in state $s$, $\mu_0$ is the initial state distribution, $R : \mathcal{S} \times \mathcal{A} \to \mathbb{R}$ is the reward function and $\gamma \in (0, 1)$ is the discount factor. A policy $\pi : \mathcal{S} \times \mathcal{A} \to \Omega(\mathcal{A})^2$ is a map from states $s \in \mathcal{S}$ to distributions $\pi(\cdot|s)$ over actions, with $\pi(a|s)$ being the probability of taking action $a$ in state $s$.

In absence of a given ground truth reward function, inverse reinforcement learning methods aim to estimate a suitable reward function based on a dataset of expert trajectories $\mathcal{D}_E = \{\xi_i\}_{i \leq K}$ where $\xi_i = (s_{1:T_i}^{(i)}, a_{1:T_i}^{(i)})$ is a sequence of states and actions of expert $i$ of length $T_i$. To achieve this goal, a number of methods based on distribution matching may be used. Such methods typically minimize a divergence measure (Csiszár, 1972) between the expert trajectories and the trajectories induced by a policy optimizing the estimated reward.

We begin by presenting maximum entropy IRL (MaxEntIRL) (Ziebart et al., 2008), which serves as a foundation for most of these methods. In MaxEntIRL, a feature matching approach is used to learn a reward

---

[2]$\Omega(\mathcal{A})$ denotes the set of probability measures over the action space $\mathcal{A}$

function $r_{\psi,\varphi} = \psi^T \varphi(s)$, where the state features $\varphi(s)$ may be specified a priori or learned using a using a neural network model (Wulfmeier et al., 2015). The policy is trained by optimizing the expected cumulative reward using the reward function estimate. The model describes the fact that the expert trajectories are sampled from a Gibbs distribution defined over trajectories:

$$p(\xi|\psi,\varphi) = \frac{1}{Z_{\varphi,\psi}} \exp(r_{\psi,\varphi}(\xi))) \tag{1}$$

which corresponds to the solution of the entropy maximization problem of the trajectory distribution under feature matching and normalization constraints.

In the more general case of stochastic dynamics $p(s_{t+1}|s_t, a_t)$ and random initial state distribution $\mu_0$, we can define the *generative model of trajectories* $p(\xi|O_{1:T})$ conditioned on the optimality variables $\{O_t\}_{t=1:T}$ as follows (Levine, 2018):

$$p(\xi|O_{1:T}) \propto p(\xi, O_{1:T}) = \mu_0(s_0) \prod_{t=1}^{T} p(O_t = 1|s_t, a_t) p(s_{t+1}|s_t, a_t) \tag{2}$$

where the conditional distribution $p(O_t = 1|s_t, a_t) \propto \exp(r_{\psi,\varphi}(s_t, a_t))$ encodes the optimality of a single timestep of the trajectory, i.e. $O_t = 1$ corresponds to an expert-level transition. The optimal reward weight solution $\psi^*$ can be obtained by maximizing the likelihood of eq. (2) w.r.t. the parameter $\psi$. We state the *primal maximum-likelihood objective* (Ziebart et al., 2008):

$$\max_{\psi} \mathbb{E}_{\xi \sim \mathcal{D}_E} \left[ \log \frac{1}{Z(\psi)} e^{\psi^T \varphi(\xi)} \right] = \max_{\psi} \mathbb{E}_{\xi \sim \mathcal{D}_E} [\psi^T \varphi(\xi) - \log Z(\psi)] \tag{3}$$

**Variational dual formulation.** Due to the difficulties of computing the log-partition function in high-dimensional spaces, a dual formulation of the maximum likelihood problem is derived. The Gibbs distribution over trajectories $p(\xi|\psi,\varphi)$ obtained via the maximum entropy principle belongs to the exponential family of distributions:

$$p(\xi|\psi,\varphi) = p_0(\xi) \exp(\psi^T \varphi(\xi) - A(\psi)) \tag{4}$$

where $A(\psi) = \log Z_{\psi} = \log \int_{\Xi} \exp(\psi^T \varphi(\xi)) p_0(\xi) d\xi$ is the log-partition function defined over the space of trajectories $\Xi$ and $p_0$ is the base measure. Leveraging the strict concavity of the log-partition function $A(\psi)$ in $\psi$ (Wainwright et al., 2008), the Fenchel-Legendre dual (Rockafellar, 2015) of $A(\psi)$ is given by

$$A(\psi) = \max_{\varphi \in \Phi} \langle \psi, \varphi(\xi) \rangle - D_{KL}(p(\xi|\psi,\varphi)||p_0(\xi)) = \max_{q \in \mathcal{P}} \langle q(\xi), g_{\psi,\varphi}(\xi) \rangle - D_{KL}(q(\xi)||p_0(\xi)) \tag{5}$$

Here, we use the generalization of the marginal polytope $\Phi$ over first-order statistics $\varphi(\xi)$ to the space of sampling distributions with bounded $L_2$-norm $\mathcal{P}$ (Dai et al., 2019) and $g_{\psi,\varphi}(\xi) = \psi^T \varphi(\xi)$. $q(\xi)$ describes a trajectory sampling distribution. In our case, the base measure $p_0$ corresponds to the Lebesgue or count measure according to the continuity of the state space, i.e. $p_0(\xi) = 1$. Thus, $D_{KL}(q||p_0)$ simplifies to the negative entropy of the sampling distribution $\mathcal{H}(q)$. By plugging in the resulting dual of the log-partition in eq. (5) into the maximum likelihood objective in eq. (3), we obtain the dual saddle-point objective:

$$\max_{\psi,\varphi} \min_{q \in \mathcal{P}} \mathbb{E}_{\xi \sim \mathcal{D}_E}[g_{\psi,\varphi}(\xi)] - \mathbb{E}_{\xi \sim q}[g_{\psi,\varphi}(\xi)] - \mathcal{H}(q) \tag{6}$$

**Connection to $f$-divergences.** The resulting dual problem is closely related to $f$-divergence (Csiszár, 1972) minimization which allows us to obtain a numerical solution strategy for eq. (6). It is well known that the maximum-likelihood problem (eq. (3)) is asymptotically equivalent to the minimization of the KL-divergence $D_{KL}(p(\xi|\theta^*)||p(\xi|\theta))$ (Andersen, 1970), where $\theta^*$ is the parameter vector maximizing the likelihood of $p(x|\theta^*)$. The variational formulation of the more general $f$-divergence minimization problem, of which $D_{KL}$ is an instance, is given by:

$$D_f(p||q) = \mathbb{E}_q \left[ f \left( \frac{p}{q} \right) \right] \geq \sup_{g:\mathcal{X} \to \mathbb{R}} \mathbb{E}_p[g(\xi)] - \mathbb{E}_q[f^*(g(\xi))] \tag{7}$$

where $f$ is a convex function and $f^*$ denotes its Fenchel conjugate. In particular, for $f(x) = \frac{1}{2}|x-1|$, we obtain the *total variation distance* $D_{TV}(P, Q)$:

$$D_{TV}(p||q) = \sup_{||g||_\infty \leq 1} \mathbb{E}_p[g(\xi)] - \mathbb{E}_q[g(\xi)] \tag{8}$$

which is equivalent to eq. (6) for a restricted class of functions $g \in \{g : \mathcal{X} \to \mathbb{R}, ||g||_\infty \leq 1\}$ and an entropy regularization term for the sampling distribution $q$.

In order to perform the minimization of the $f$-divergence objective in eq. (7), we can leverage a correspondence between optimal risk functions and $f$-divergences. In particular, for a given $f$-divergence $D_f$, there exists a corresponding decreasing convex risk function $\phi(\alpha)$ of the classification margin $\alpha$, such that the optimal risk $R_\phi = -D_f$(Nguyen et al., 2009, Thm. 1). This correspondence allows the objective to be described as a two-player zero-sum game and is amenable to optimization using *generative-adversarial-network*-like (Goodfellow et al., 2014; Finn et al., 2016a) frameworks. When used to minimize the divergence between the expert and policy occupancy measures, this problem class includes many adversarial imitation learning algorithms (Ho & Ermon, 2016; Fu et al., 2017; Ni et al., 2021). We will use these algorithms as solution strategies for eq. (6).

Now that we have both exact and approximate formulations of the IRL problem (eq. (3), eq. (6)) we shall see how spurious correlations can arise when the expectations over the datasets $\mathcal{D}_E$ in eq. (3) and eq. (6) are evaluated over samples from diverse sources.

## 2.2   Spurious correlations and causal invariance approaches

We shall now outline the intuition as to what we consider spurious correlations in inverse reinforcement learning. It is necessary, at this point, to introduce structural causal models, which will help define the notion of spurious correlations. A structural causal model (SCM) (Pearl, 2009) is defined as a tuple $\mathfrak{G} = (S, P(\varepsilon))$, where $P(\varepsilon) = \prod_{i \leq K} P(\varepsilon_i)$ is a product distribution over exogenous latent variables $\varepsilon_i$ and $S = \{f_1, ..., f_K\}$ is a set of structural mechanisms where $\mathrm{pa}(i)$ denotes the set of parent nodes of variable $x_i$: $x_i := f_i(\mathrm{pa}(x_i), \varepsilon_i)$ for $i \in |S|$. $\mathfrak{G}$ induces a directed acyclic graph (DAG) over the variables nodes $x_i$. The SCM entails a joint observational distribution $P_\mathfrak{G} = \prod_{i \leq K} p(x_i|\mathrm{pa}(x_i))$ over variables $x_i$ conditioned on the parents of $x_i$ for some probability distribution $p(\cdot|\mathrm{pa}(x_i))$ describing the mechanism $f_i$. Interventions on $\mathfrak{G}$ constitute modifications of structural mechanisms $f_i$ yielding interventional distributions $\tilde{P}_\mathfrak{G}$. In the context of IRL, we consider interventional settings of the expert trajectory distribution $p(\xi|\psi, \varphi)$ in eq. (2).

In supervised learning, a correlation between the input representation and the label is considered spurious if it does not generalize under distribution shift, e.g. when the trained model is evaluated on a test set of examples sampled out-of-distribution. More specifically, the distribution shift constitutes an intervention on the causal parents of the target label, which allows the application of invariance based approaches, that we will describe below. In the case of IRL, we consider a correlation to be spurious when a reward function does not generalize to perturbations in the initial measure or dynamics of the MDP, i.e. the policy optimizing a reward that reflects this correlation will absorb this signal and fail to perform optimally under perturbed environment dynamics.

**Transition SCM.**   To illustrate scenarios in which such spurious correlations arise, we consider the structural causal model of a transition in Figure 1(a). At timestep $t$, the model is composed of the state $s_t$, action $a_t$, next state $s_{t+1}$, the optimality label $O_t$ describing the optimality of the transition at timestep $t$, a variable $E$ encoding the setting index $e \in I(\mathcal{E}_{tr})$ and an unobserved latent variable $C$, which might vary as $E$ changes. In this model, we would like to avoid non-causal information paths between the environment index $E$ and the optimality label. The presence of such paths corresponds to spurious correlations. We will now discuss specific scenarios in which such correlations arise and subsequently describe a way to mitigate them.

In Figure 1b, we observe a general partitioning of an arbitrary transition input $(s, a, s')$ into the *causal* transition feature components $\mathbf{x}^{(c)}$ and *non-causal* transition feature components $\mathbf{x}^{(nc)}$ described by feature function $\varphi : \mathcal{S} \times \mathcal{A} \times \mathcal{S} \to \mathbb{R}^d$. Here, conditioning on the $\mathbf{x}^{(nc)}$ *collider* introduces a spurious correlation path. Among other causes, this situation can arise due to selection bias caused by a prevalence of certain transitions

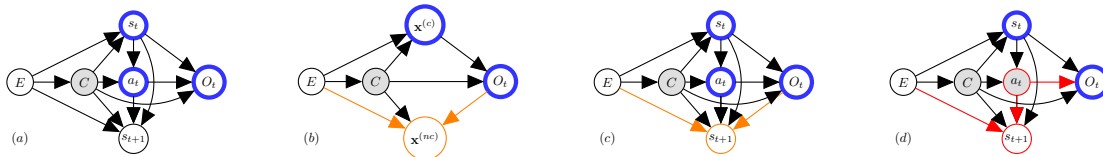

Figure 1: (a) Probabilistic graphical model of a transition under influence of the index variable $E$ and latent variable $C$. The invariant conditional is highlighted in blue. (b) General setting where $O_t$ depends on causal $\mathbf{x}^{(c)}$ and non-causal $\mathbf{x}^{(nc)}$ features of the transition. Conditioning on the collider variable (in orange) creates a spurious correlation path. (c) Collider conditioning assuming wrong edge orientation $O_t \to s_{t+1}$. This corresponds to $O_t$ being the causal parent of $s_{t+1}$ (d) Spurious correlations arising under assumption of state-only formulation of the reward. Since $a_t$ is unobserved, a backdoor path (in red) is formed.

in the pooled dataset, i.e., the pooling of *diverse sources of demonstrations* under the binary optimality label introduces a spurious correlation between states visited by expert policies as a consequence of the respective preferences and the binary optimality label.

A second scenario can be observed in Figure 1c. This scenario requires the assumption that the orientation of the edge from node $O_t$ to node $s_{t+1}$ is temporally causal, meaning that the optimality of a state $s_t$ at time $t$ is a causal parent of the next state. In this case, observing the collider node $s_{t+1}$ makes nodes $E$ and $O_t$ conditionally dependent [3]: $E \not\perp\!\!\!\perp O_t|s_{t+1}$. In Figure 1d we can observe the scenario where we do not condition the representation of the optimality conditional on the action, which corresponds to the learning from observations modality (Zhu et al., 2020). By conditioning on the collider node $s_{t+1}$ and not observing the action node $a$, a *backdoor path* (Pearl, 2009) is formed between the setting index $E$ and the optimality variable $O_t$, resulting in the violation of their conditional independence relationship.

**Dataset partitioning and training settings.** The standard IRL procedure typically pools input demonstration data into one dataset which may lead to absorption of these correlations by the learned reward. To resolve this, we first need to make the assumption that the data is partitioned according to different sources obtained from observational and interventional settings of the trajectory distribution. This assumption is valid in three different scenarios: (i) expert demonstrations were gathered on different environment dynamics, (ii) initial states were sampled from different initial state distributions or (iii) experts have reward preferences, i.e. optimize a perturbed version of the reward associated with the true task goal. In the context of the current work, we consider the source diversity to be a consequence of the individual expert preferences expressed as reward functions parameterizing eq. (1). We denote the union set of samples obtained from these settings as $\mathcal{E}_{tr}$, the set of *training settings*, and assume access to multiple such settings $\mathcal{D}_e := \{(\xi_i)\}_{i=1}^{|\mathcal{E}_{tr}|}$ during training.

**From robustness to invariance.** The consideration of multiple source distributions and associated target noise warrants the application of robust methods such as distributionally robust optimization (DRO) (Namkoong & Duchi, 2016; Bashiri et al., 2021; Viano et al., 2021). Such approaches modify the loss objective by searching over the space of empirical distributions indexed by $e \in \mathcal{E}_{tr}$ under which the expected loss $\mathcal{L}_e$ describing the problem objective is maximized. This is implemented in practice by searching over the set of training settings $\mathcal{E}_{tr}$ resulting in a min-max objective for some function class $f \in \mathcal{F}$:

$$\min_{f \in \mathcal{F}} \max_{e \in \mathcal{E}_{tr}} \mathbb{E}_{\xi \sim \mathcal{D}_e} \mathcal{L}_e(f, \xi) \tag{9}$$

This effectively regularizes the model by optimizing based on its *worst-case* performance. While this modification can tackle the issue of model overfitting to scarce data, it does not address potential diversity of the data due to mode-seeking behaviour of the min-max problem (Rahimian & Mehrotra, 2019). This behaviour describes the fact that the maximum "latches on" to the training setting with the largest likelihood loss which might constitute a spurious mode of the demonstrations with respect to the actual task goal. Arjovsky et al. show that the solution of robust regression is a first order stationary point of the weighted square error (in the case of convex losses), if the variance of the loss is used as a per-setting bias (Arjovsky et al., 2019, Prop. 2). This limits the generalization capacity to the convex hull of the training settings $\mathcal{E}_{tr}$.

---

[3]Here, $\perp\!\!\!\perp$ denotes statistical independence

Following (Arjovsky et al., 2019), we consider causal properties of the robust estimation problem for the purposes of improved generalization outside of the convex hull of training settings. The *causal invariance principle* postulates that for the problem of estimating a target conditional, e.g. classification label, one should only consider variables which belong to the set of causal parents of the target. More specifically, causally invariant covariates must yield the same conditional distribution of the target in both observational and interventional settings of the SCM. Lifting the distributional restrictions of the methods described in (Peters et al., 2015), the invariant risk minimization (IRM) principle (Arjovsky et al., 2019) aims to identify a causally invariant data representation $\varphi$ by instantiating a bi-level optimization problem for a representation function $\varphi$ and a predictor function $w$:

$$\min_{\varphi:\mathcal{X}\to\mathcal{H},w:\mathcal{H}\to\mathcal{Y}} \sum_{e\in\mathcal{E}_{tr}} \mathcal{L}^e(w\circ\varphi) \ \ \text{s.t.:} \ \ w\in\operatorname*{argmin}_{\bar{w}:\mathcal{H}\to\mathcal{Y}} \mathcal{L}^e(\bar{w}\circ\varphi) \quad \forall e\in\mathcal{E}_{tr} \tag{10}$$

This objective admits an unconstrained relaxation using the gradient norm penalty $\mathbb{D}(w = 1.0, \varphi, e) = ||\nabla_{w|w=1.0}\mathcal{L}^e(w\circ\varphi)||^2$ which quantifies the optimality of a fixed predictor ($w = 1.0$) at each setting $e$ (Arjovsky et al., 2019). This leads to the following unconstrained formulation of the learning problem:

$$\min_{\varphi:\mathcal{X}\to\mathcal{Y}} \sum_{e\in\mathcal{E}_{tr}} \mathcal{L}^e(\varphi) + \lambda||\nabla_{w|w=1.0}\mathcal{L}^e(w\circ\varphi)||^2 \tag{11}$$

In the following section, we will show how to incorporate this principle into the IRL setting.

## 2.3 Reward regularization using causal invariance

Mapping the objective in eq. (10) to the context of reward learning, we consider data gathered by different experts to correspond to interventional settings of the trajectory distribution in eq. (2), where the interventions reflect the varying preferences exhibited by the policy of the respective experts. The stable conditional we would like to identify corresponds to the conditional distribution of the optimality label $P(O_t|s_t, a_t)$. To do so, we invoke the causal invariance principle to learn reward functions which utilize features which are invariant to some class of deviations exhibited by the experts. We motivate this by the fact that despite the discrepancies in the demonstrations, all experts are assumed to perform the task in an optimal fashion with respect to the true task goal. This implies that all experts, at least in part, optimize the same underlying reward that we would like to recover. In doing so, we hope to extract succinct descriptions of the underlying agreed intentions of the experts as reward functions. As a result, we expect such rewards to be more readily applicable to MDPs with distribution shift of the dynamics. As a next step, we show how to apply this principle into practice by introducing the causal invariance (CI) regularization, instantiated as the IRM penalty described in eq. (11).

**Feature matching regularization.** We begin by considering the maximum entropy feature matching problem. For the Gibbs distribution $p(\xi|\psi, \varphi) = \exp(\psi^T\varphi(\xi))/Z_{\varphi,\psi}$ over trajectories, we can write down the constrained optimization problem analogously to eq. (10):

$$\max_{\varphi,\psi} \sum_{e\in\mathcal{E}_{tr}} \sum_{\xi\in\mathcal{D}_e} \log p(\xi|\psi, \varphi) \ \text{s.t.} \ \psi\in\operatorname*{argmax}_{\bar{\psi}} \sum_{\xi\in\mathcal{D}_e} \log p(\xi|\bar{\psi}, \varphi) \tag{12}$$

Intuitively, due to convexity of the likelihood function w.r.t. the natural parameter $\psi$ (Wainwright et al., 2008), this corresponds to penalizing the deviation $\psi$ in $p(\xi|\psi, \varphi)$ from the optimal parameter $\psi^*$ which maximizes the likelihood $p(\xi|\psi^*, \varphi)$.

In an analogous fashion to the IRM approach described above (eq. (10)), we propose to relax the constrained optimization problem by defining a regularization term $\mathbb{D}(\psi, \varphi, e)$ which describes this deviation. Here, the expected loss $\mathcal{L}^e$ corresponds to the primal maximum likelihood loss of the Gibbs distribution over trajectories.

**Definition 1.** *Let $\mathcal{E}_{tr}$ be the set of training settings and $\psi, \varphi$ be the parameters of the likelihood $p(\xi|\psi, \varphi)$. $\mathbb{D}(\psi, \varphi, e)$ is a distance function representing the violation of the constraints of eq.* (12) *in training setting $e\in\mathcal{E}_{tr}$ w.r.t. the optimal solution.*

$$\mathbb{D}(\psi, \varphi; e) = ||\nabla_{\psi|\psi=1.0}\mathcal{L}^e(\psi, \varphi)||^2 \tag{13}$$

---

**Algorithm 1** CI regularized Feature Matching IRL (CI-FMIRL)

---

**Input:** Expert trajectories $\mathcal{D}_E^e$ assumed to be obtained from multiple experts *by intervening on $p(\xi|\psi,\varphi)$*
**Init:** Initialize reward estimate $r_\psi$ and state feature network $\varphi_\theta$
**for** setting $e$ in $\{1,...,\mathcal{E}_{tr}\}$ **do**
    **while** $r_{\psi,\varphi}$ not converged **do**
        Compute feature matching gradient $\nabla_\psi \mathcal{L}(\psi,\varphi;e) = \mathbb{E}_{\mathcal{D}_E^e}[\varphi(\xi)] - \mathbb{E}_{p(\xi|\psi)}[\varphi(\xi)]$ and *causal invariance*
        penalty gradient $\nabla_\varphi \mathbb{D}(\psi,\varphi;e)$ and backpropagate the weighted sum through feature network $\varphi_\theta(s)$
        Compute policy $\pi_{r_{\psi,\varphi}}$ using value iteration on the reward estimate $r_{\psi,\varphi}$
    **end for**
**end for**
**Return:** Trained reward $r_{\psi,\varphi}$

---

In our case, the deviation is computed w.r.t. the parameters maximizing the likelihood of the Gibbs distribution due to convexity of $\mathcal{L}_{MLE}$ w.r.t. function $g_{\psi,\varphi}$. Applying the gradient of this penalty to the representation function $\varphi$ effectively regularizes the representation to minimize the constraint violations. In simple tabular MDPs, where the computation of the partition function is tractable, we directly apply this regularizer to the primal maximum likelihood objective as follows:

$$\mathcal{L}_{MLE}(\psi,\varphi,e) = \max_{\varphi,\psi} \sum_{e \in \mathcal{E}_{tr}} \left( \mathbb{E}_{\xi \in \mathcal{D}_e} \left[ \log \left( \frac{1}{Z_{\psi,\varphi}} \exp(\psi^T \varphi(\xi)) \right) + \lambda \mathbb{D}(\psi,\varphi,e) \right] \right) \tag{14}$$

In the primal case, for a trajectory distribution described by an exponential family, we can derive a closed form of the gradient penalty. We summarize this result as the following proposition [4]:

**Proposition 1.** *Let the likelihood $p(\xi)$ belong to a natural exponential family with parameter $\psi$, sufficient statistics $\varphi(x)$ and the (Lebesgue) base measure $p_0$. Let $\mathcal{D}_E^e$ be the dataset corresponding to interventional setting $e$. Then, for all $e \in \mathcal{E}_{tr}$, the causal invariance penalty for the maximum likelihood loss is the norm of the sufficient statistics expectation difference:*

$$\mathbb{D}(\psi,\varphi;e) = ||\nabla_{\psi|\psi=1.0} \mathcal{L}^e(\psi,\varphi)||^2 = ||\mathbb{E}_{\mathcal{D}_E^e}[\varphi(\xi)] - \mathbb{E}_{p(\xi|\psi)}[\varphi(\xi)]||^2 \tag{15}$$

This closed form of the gradient norm penalty can be utilized in the maximum causal entropy solver (Ziebart et al., 2010). We assume the state features to be the output of a neural network according to the DEEPMAXENT model (Wulfmeier et al., 2015). In order for the network to adopt invariant features, the gradient norm penalty in eq. (15) is used to update the feature network using backpropagation [5]. The resulting algorithm (CI-FMIRL) is presented in Algorithm 1.

**Penalty for dual formulation.** In large scale MDPs, where the evaluation of the log-partition function is intractable, we use the variational dual objective outlined in eq. (5). We will now show how to apply the same regularization to the variational dual formulation of the problem outlined in eq. (5). In order to do so, we first need to derive the causal invariance penalty for the dual formulation. By leveraging the strict concavity of *eq. (6)* w.r.t. *q*, we can straightforwardly extend the distance penalty to the dual formulation as follows:

$$\mathcal{L}_{dual}(\psi,\varphi,q,e) = \max_{\psi,\varphi} \sum_{e \in \mathcal{E}_{tr}} \min_q \left[ \mathbb{E}_{\xi \sim \mathcal{D}_E^e}[g_{\psi,\varphi}(\xi)] - \mathbb{E}_{\xi \sim q}[g_{\psi,\varphi}(\xi)] + \mathcal{H}(q) \right] + \lambda \mathbb{D}(\psi,\varphi,e) \tag{16}$$

The numerical solution strategy for the saddle-point objective in Equation (16) can be realized using a two-player min-max game implemented using a GAN-like framework (Finn et al., 2016b). More specifically, an equivalence between the maximum likelihood problem for a Gibbs distribution over trajectories and the

---

[4]All omitted proofs are found in appendix A
[5]We derive a closed form of the gradient estimate in Appendix D

---

**Algorithm 2** CI regularized Adversarial IRL (CI-AIRL)

---

**Input:** Expert trajectories $\mathcal{D}_E^e$ assumed to be obtained from multiple experts *by intervening on* $p(\xi|\psi, \varphi)$
**Init:** Initialize actor-critic $\pi_\theta, \nu_\vartheta$ and discriminator $g_{\xi,\varphi}$
**for** setting $e$ in $\{1, ..., \mathcal{E}_{tr}\}$ **do**
    Collect trajectory buffer $\mathcal{D}_\pi = \{\xi_i\}_{i \leq |\mathcal{D}_\pi|}$ by executing the policy $\pi_\theta$
    Update $g_{\varphi,\theta}(s, a)$ via binary logistic regression by maximizing

$$\mathcal{L}(\varphi, \psi; e) = \mathcal{L}_{\text{BCE}}(\xi, \varphi, \psi; e) + \lambda||\nabla_{\psi|\psi=1.0}\mathcal{L}_{\text{BCE}}(\xi, \varphi, \psi; e)||^2$$

    using dataset tuple $(\mathcal{D}_E^e, \mathcal{D}_\pi)$
    Update actor-critic $(\pi_\theta, \nu_\vartheta)$ w.r.t. the reward function of the *regularized discriminator* using the *soft-actor-critic* RL procedure
**end for**
**Return:** Trained reward $r_{\varphi,\psi}$ and actor-critic $\pi_\theta, \nu_\vartheta$

---

corresponding variational approximation has been shown in (Fu et al., 2017, App. A). We state the CI gradient penalty as a result of the following proposition.

**Proposition 2.** *The gradient of the primal exponential family maximum-likelihood problem in Equation* (3) *w.r.t. the natural parameter* $\psi$ *is equivalent to the gradient of the dual in Equation* (5) *w.r.t the parameter* $\psi$ *when the density ratio* $\frac{q}{p}$ *is unity.*

$$||\nabla_\psi\mathcal{L}_{dual}(\psi, \varphi, q, e)||_2 = ||\min_q \mathbb{E}_{\xi\sim\mathcal{D}_E}[\varphi(\xi)] - \mathbb{E}_{\xi\sim p(\xi|\psi,\varphi)}\left[\frac{q(\xi)}{p(\xi|\psi,\varphi)}\varphi(\xi)\right]||_2$$

Using this result, we can apply the gradient penalty to the dual. Effectively, the resulting gradient estimate requires an importance sampling estimate of the second expectation using the sampling trajectory distribution $q$ induced by the policy. The minimum over $q$ is attained when the importance weight is unity.

As we have seen in section 2.1, the dual objective is closely related to the $f$-divergence minimization objectives which form the basis of multiple *adversarial imitation learning* algorithms (Ho & Ermon, 2016; Fu et al., 2017; Ni et al., 2021). This class of algorithms leverages the correspondence between the $f$-divergence objectives and equivalent binary classification surrogate losses as described in (Nguyen et al., 2009). One such example is the optimal logistic loss of a binary classification function $g_D(x) = \frac{p(x)}{p(x)+q(x)}$ which corresponds to the Jensen-Shannon divergence (Ho & Ermon, 2016) between distributions $p(x)$ and $q(x)$:

$$\max_{g_D} \mathcal{L}_{BCE}(g_D) = \max_g \mathbb{E}_{x\sim\mathcal{D}_E}[\log g_D(x)] + \mathbb{E}_{x\sim\pi}[\log(1 - g_D(x))] = D_{JS}(p||q) \tag{17}$$

In practice, we can make use of this equivalence in order to apply the penalty defined in definition 1 solely to the discriminator function $g_D(x)$ which classifies transitions sampled from the expert datasets $\mathcal{D}_E^e$ and transitions obtained from the imitation policy.

**Algorithm description.** We present the resulting adversarial algorithm (CI-AIRL) in Algorithm 2. The algorithm describes a two-player zero-sum game between an agent parameterized using a soft-actor-critic architecture and the discriminator which is used to provide a reward function by distinguishing between expert and policy samples. The adversarial training procedure generally mimics that of divergence-based methods such as (Ho & Ermon, 2016; Fu et al., 2017). There are three main differences compared to baseline adversarial training algorithms. The first is the fact that we use multiple experts in a distinct fashion as opposed to pooling the demonstrations into one big dataset. The second is the regularization of the discriminator objective in Equation (17) using the gradient norm penalty $\mathbb{D}(\xi, \varphi, \psi; e) = ||\nabla_{\psi|\psi=1.0}\mathcal{L}_{\text{BCE}}(\xi, \varphi, \psi; e)||^2$ in a similar fashion to eq. (10), where $\psi = 1.0$ corresponds to a fixed scalar predictor. Finally, we utilize soft-actor-critic (SAC) (Haarnoja et al., 2018) as an instance of an off-policy forward RL solution method as opposed to the on-policy algorithms used in (Ho & Ermon, 2016; Fu et al., 2017).

# 3 Experiments

We will now evaluate the proposed method empirically. The experiments are designed to answer the following questions: (i) What is the effect of the causal invariance penalty on the recovered reward structure? (ii) Does regularizing the reward function using causal invariance improve downstream policy performance? (iii) What is the impact of changing the loss function of the discriminator and (iv) increasing the perturbation magnitude? To answer these questions, we evaluate our model in two settings.

The first setting considers a tabular gridworld scenario, where the partition function is tractable. We perform reward learning experiments using variants of the maximum entropy feature expectation matching algorithm. In particular, we use the DEEPMAXENT (Wulfmeier et al., 2015) model of state features with different regularization strategies. In the second setting, we test the invariance regularization in an adversarial IRL setting on simulated robotic locomotion environments. Here, we demonstrate the generalization of the obtained reward functions by retraining policies using the recovered rewards on perturbed versions of the environment dynamics.

## 3.1 Tractable setting: Gridworld experiments using feature matching

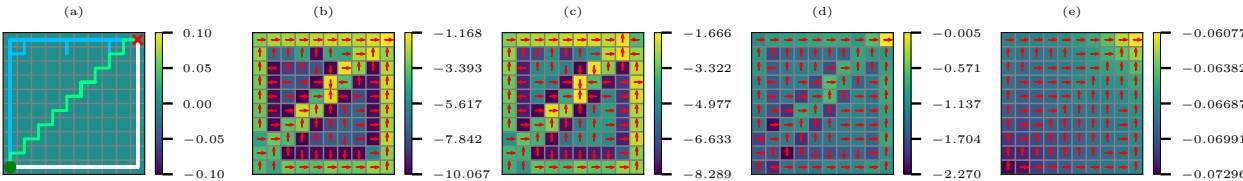

Figure 2: Feature matching reward recovery on a gridworld environment. (a) expert trajectory datasets: 1st group (blue) 400 trajectories, 2nd group (white): 25 trajectories, 3rd group (green): 3 trajectories. (b) MaxEnt IRL ERM baseline (c) MaxEnt IRL ERM baseline with L2 regularization coeffcient $\lambda_{L2} = 1e-3$ (d) MaxEnt IRL with CI penalty, $\lambda_I = 0.01$, (e) MaxEnt IRL with CI penalty, $\lambda_I = 0.05$

For the first experiment, we aim to illustrate the principle of causal invariance regularization in the tractable IRL setting using Algorithm 1. To do so, we choose a simple gridworld environment with stochastic dynamics and a sparse ground truth reward structure and corresponding trajectories illustrated in (Figure 2a). The goal of the agent is to navigate from the bottom left to the top right corner. Due to the tabular nature of the state space, this setting allows a direct visual comparison of the recovered reward functions for the different regularization strategies.

**Setup.** In order to construct the dataset settings $\mathcal{E}_{tr}$, we generate a dataset of 3 groups of expert trajectories using a value iteration method on modified versions of the MDP. The initial and final states of the trajectories are fixed. We introduce a selection bias into the IRL feature expectation matching problem by manipulating the expert preferences to choose different paths. This results in a trajectory dataset with different number of trajectories for each of the three paths chosen by the experts (Figure 2a): 40 trajectories for 1st group, 10 trajectories for 2nd group and 1 trajectory for the 3rd group.

**Baselines.** Throughout this section, we compare the proposed regularization to both non-regularized and regularized versions of the DEEPMAXENT algorithm. In particular, we use an L2 penalties of the reward feature weights $\varphi(s)$ and a Lipschitz smoothness penalty (Yoshida & Miyato, 2017) as baseline regularization strategies.

**Results.** In Figure 2, we can observe that both the unregularized MaxEnt IRL algorithm (ERM) (Figure 2b) and $L_2$-regularized MaxEnt IRL algorithm (ERM-$L_2$) (Figure 2c) exhibit overfitting to the expert datasets and partially fail to recover a meaningful reward and respective policy. In contrast, the IRM-regularized version recovers a shaped reward function which takes the different optimal paths into account in a manner which demonstrates an invariance to the setting index $e$. In particular, increasing the regularization strength $\lambda$ improves the reward significantly (Figure 2d - Figure 2e). It is easy to see that the CI-regularized reward can more straightforwardly be used in a setting where the dynamics of the MDP might be modified, e.g. when obstacles are introduced.

## 3.2 Adversarial setting

For the second experiment, we perform experiments in large state spaces, which require the use of Algorithm 2. Our primary goal is to investigate how well the recovered reward function allows the elicitation of a policy under change of dynamics. To do so, we first learn a reward function using the adversarial training procedure and then retrain a policy from scratch using the recovered reward as training signal.

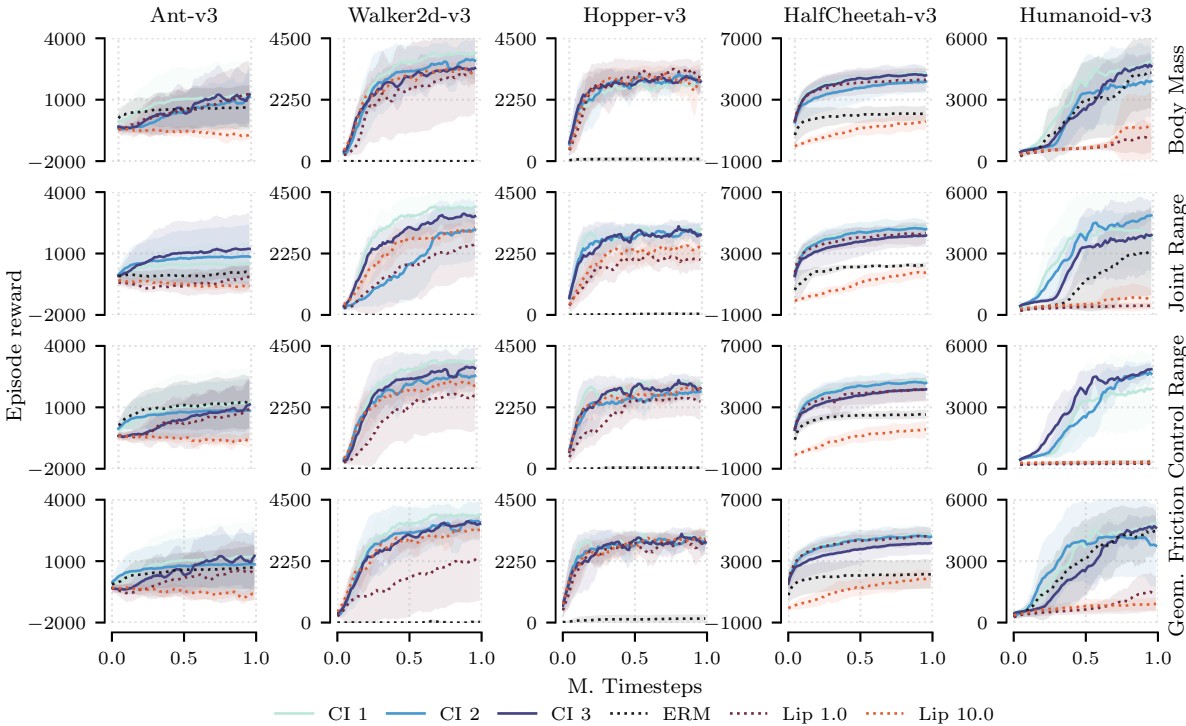

Figure 3: Comparison of SAC policy performance w.r.t. ground truth reward when trained on inferred reward functions. Every row depicts a different type of dynamics perturbation for the five MuJoCo tasks as described in section 3.2. Here, AIRL is chosen as the baseline algorithm. The variants correspond to the unregularized baseline: ERM, Lipschitz regularization: LIP and three best CI regularization parameters CI.

**Setup.** For our experimental setting, we choose a set of robot locomotion tasks from the MuJoCo (Todorov et al., 2012) suite. We generate the demonstration datasets by using pretrained soft-actor-critic (SAC) policies from the STABLE-BASELINES3 repository [6]. In order to diversify the demonstrations, we perturb the policies using a structured noise approach: the optimal policy action is perturbed with Gaussian noise at every timestep with a probability $p = 0.3$ of the noise being applied. We have used 10 expert trajectories for every environment in all the experiments performed in this section. The reward function is obtained by using a number of different discriminator functions corresponding to variations of the $f$-divergence objective. In order to assess the quality of the recovered reward, we retrain policies on the recovered reward under distribution shift of the dynamics realized by perturbing physical parameters of the simulation. Specifically, we apply Gaussian noise to four parameters of the MuJoCo simulation: the body mass, joint range and actuator control range of the robot as well as the contact friction coefficient of the simulated surface. We evaluate the behaviour of the algorithms for a variety of perturbation magnitudes.[7]

**Baselines.** Throughout this section, we use three different adversarial IRL algorithms as baselines for reward learning: (i) AIRL (Fu et al., 2017), an adversarial IRL approach which relies on a structured discriminator to recover a stationary reward function (ii) MEIRL, an adaptation of the maximum entropy IRL algorithm for large state spaces *without* an importance sampling estimator (Ni et al., 2021) and (iii) GAIL (Ho & Ermon,

---

[6]https://huggingface.co/sb3

[7]Additional experimental details are provided in appendix C

2016), an imitation learning where we extract the unshaped discriminator as a reward function for policy learning. All baselines use the SAC (Haarnoja et al., 2018) algorithm as a forward RL agent for purposes of sample efficiency.

**Results.** Figure 3 depicts the results of a *SAC* agent trained using the reward function recovered by the AIRL baseline algorithm and two regularization strategies: (i) the Lipschitz smoothness regularizer (Gulrajani et al., 2017) controlled by the penalty coefficient $\lambda_L$ and the proposed regularization in definition 1, controlled by the penalty coefficient $\lambda_I$. We evaluate the models on combinations of regularization coefficients drawn from the following sets: $\lambda_I \in \{0.0, 0.1, 1.0, 10.0, 100.0\}$ and $\lambda_L \in \{0.0, 1.0, 10.0\}$ and pick the three best performing $\lambda_I$ coefficients for every environment. We can observe that applying the causal invariance penalty leads to superior performance compared to using non-regularized or Lipschitz regularized rewards. The choice of the $\lambda_I$ hyperparameter is problem dependent – we report three best performing variants per-environment.

**Impact of adversarial algorithm variation.** We evaluate our method on a number of variants of the adversarial imitation learning algorithms based on $f$-divergence minimization, as outlined above. The results are presented in table 1. We report the ground truth reward performance attained using an SAC agent trained using the recovered rewards after 1 million timesteps. The regularization coefficients are selected on a per-environment basis. The full results tables are provided in appendix B.4. We observe improved cumulative ground truth reward metrics for all three algorithms when compared to both the unregularized (ERM) and Lipschitz regularized (Lip) baselines. The improvement is most pronounced in the ANT, WALKER2D and HUMANOID environments, which are of higher state-space dimensionality.

**Perturbation magnitude.** In this experiment, we investigate the policy ground truth performance as a function of the perturbation magnitude applied to the physical parameters of the environment dynamics. In fig. 4, we observe that using the CI-penalty improves the ground truth episode reward performance of the trained policies for a large spectrum of perturbations for 3 out of 5 environments and does not suffer a performance penalty for the other two. [8]

## 4 Related work

**Invariance and causality in RL.** Following the introduction of causally invariant methods for supervised and representation learning (Peters et al., 2015; Heinze-Deml et al., 2017; Arjovsky et al., 2019; Ahuja et al., 2020), the concept of causal invariance has been used in a number of reinforcement learning works. Invariant causal prediction has been utilized in (Zhang et al., 2020) to learn model invariant state abstractions in a multiple MDP setting with a shared latent space. Invariant policy optimization (Sonar et al., 2021) uses the IRM games (Ahuja et al., 2020) formulation to learn policies invariant to certain domain variations. The authors of (de Haan et al., 2019) tackle the problem of causal confusion in imitation learning by making

---

[8]We provide a number of additional evaluations for both the tractable and approximate settings in appendix B

Table 1: Policy rollout results using ground truth reward for perturbed MuJoCo environments after being trained for 1M timesteps using the rewards recovered from the different discriminators in section 3.2. Here, the *body mass* parameter is perturbed with a noise magnitude of $\varepsilon = 0.2$. The results are averaged over 10 rollouts and obtained by training the model using five different random seeds.

| Environment | ANT-V3 | WALKER2D-V3 | HOPPER-V3 | HALFCHEETAH-V3 | HUMANOID-V3 |
|---|---|---|---|---|---|
| Expert | $3168.49_{\pm 1715.68}$ | $3565.33_{\pm 527.40}$ | $3119.54_{\pm 524.36}$ | $4340.61_{\pm 2020.14}$ | $4774.17_{\pm 2063.52}$ |
| AIRL (ERM) | $580.78_{\pm 1048.73}$ | $-3.29_{\pm 0.71}$ | $77.64_{\pm 88.27}$ | $2046.29_{\pm 460.98}$ | $4451.74_{\pm 1759.31}$ |
| AIRL (Lip) | $1194.04_{\pm 1583.08}$ | $3388.48_{\pm 1586.45}$ | $\mathbf{3382.91}_{\pm 234.02}$ | $4388.94_{\pm 726.69}$ | $1788.85_{\pm 1643.00}$ |
| AIRL (CI) | $\mathbf{1880.42}_{\pm 935.15}$ | $\mathbf{4162.70}_{\pm 517.13}$ | $3334.91_{\pm 221.80}$ | $\mathbf{4477.97}_{\pm 532.72}$ | $\mathbf{5107.54}_{\pm 119.31}$ |
| GAIL (ERM) | $-746.31_{\pm 468.03}$ | $328.44_{\pm 66.02}$ | $1637.50_{\pm 1419.59}$ | $886.25_{\pm 404.82}$ | $122.62_{\pm 71.53}$ |
| GAIL (Lip) | $220.97_{\pm 524.83}$ | $553.36_{\pm 277.24}$ | $1832.39_{\pm 832.32}$ | $1403.77_{\pm 1282.75}$ | $77.24_{\pm 4.66}$ |
| GAIL (CI) | $\mathbf{230.43}_{\pm 565.68}$ | $\mathbf{1172.57}_{\pm 539.86}$ | $\mathbf{2636.65}_{\pm 1114.94}$ | $\mathbf{2365.55}_{\pm 1679.64}$ | $\mathbf{549.63}_{\pm 1692.08}$ |
| MEIRL (ERM) | $-66.66_{\pm 112.03}$ | $169.11_{\pm 344.87}$ | $3.22_{\pm 0.22}$ | $-177.39_{\pm 211.43}$ | $55.99_{\pm 3.45}$ |
| MEIRL (Lip) | $-365.41_{\pm 143.70}$ | $917.14_{\pm 132.05}$ | $1045.40_{\pm 54.76}$ | $-335.10_{\pm 84.66}$ | $1001.49_{\pm 1889.60}$ |
| MEIRL (CI) | $\mathbf{153.43}_{\pm 1134.46}$ | $\mathbf{2520.24}_{\pm 994.27}$ | $\mathbf{2351.07}_{\pm 679.37}$ | $\mathbf{1371.59}_{\pm 1469.01}$ | $\mathbf{3099.51}_{\pm 2411.21}$ |

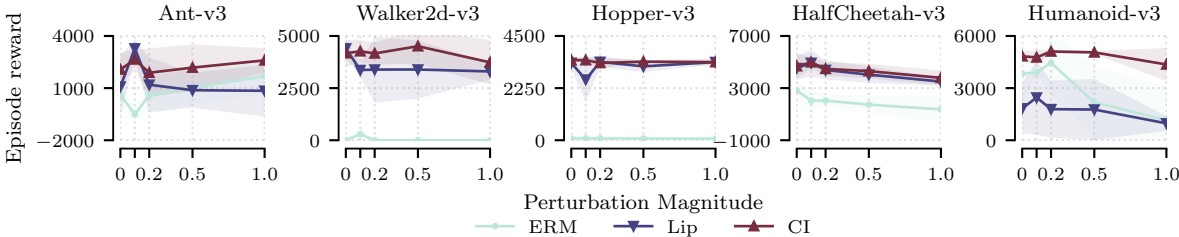

Figure 4: Comparison of SAC policy performance w.r.t. ground truth reward when trained on recovered reward functions as a function of *perturbation magnitude* of the *body mass* parameter. Here, AIRL is chosen as the baseline algorithm. ERM denotes the unregularized baseline, LIP the best Lipschitz regularization hyperparameters per environment and CI the best causal invariance regularization hyperperameters.

use of causal structure of demonstrations. Causal imitation learning under temporally correlated noise has been studied in (Swamy et al., 2022). In the offline RL setting without access to environment interactions, (Bica et al., 2021) propose to use the invariance principle for policy generalization. To the best of our knowledge, our algorithm is the first proposed method to use invariant causal prediction in the context of inverse reinforcement learning with the primary purpose focused on recovery of reward functions and their subsequent deployment for downstream purposes.

**Learning from diverse demonstrations.** Li et al. investigate the issue of imitation learning from diverse experts through the lens of identifying latent factors of variation. The authors of (Zolna et al., 2019) propose a model which also tackles the issue of spurious correlations being absorbed from expert data. However, their focus is on visual features in a solely imitation learning setting as opposed to our approach, which recovers reward functions that perform favorably in a transfer setting. Diverse demonstrations have been studied in both the context of adversarial imitation learning under assumptions on latent variables in (Tangkaratt et al., 2020) as well as offline imitation learning (Kim et al., 2021) under assumptions of explicit access to a dataset of suboptimal demonstrations. The authors of (Haug et al., 2020) propose to use suboptimal demonstrations to derive an additional supervision signal by way of matching optimality profiles for preference learning.

**Comparison to other regularization techniques.** The divergence-based dual objective used in this work admits a number of regularization strategies, which result as a the restriction of the critic function class $\mathcal{F}$. A commonly used regularization is the Lipschitz smoothness penalty (Gulrajani et al., 2017; Yoshida & Miyato, 2017) which restricts the class of functions $\mathcal{F}$ in eq. (6) to the class of Lipschitz smooth functions. Contrary to methods which penalizes the estimated gradient norm w.r.t. the input, the causal invariance penalty penalized the norm of the gradient w.r.t. to the predictor parameters of the model which vary across settings $\mathcal{E}_{tr}$. Kim & Park restrict the function class $\mathcal{F}$ to belong to an RKHS space, resulting in an imitation learning method based on the Maximum Mean Discrepancy (MMD) metric. The authors of (Bashiri et al., 2021) present a distributionally robust imitation learning method which generalizes the maximum entropy IRL robustness properties from logistic loss to arbitrary losses. In comparison, our method tackles the issue of learning rewards as opposed to imitation policy learning and allows for improved generalization due to the properties of the causal invariance penalty.

# 5 Conclusion

In this work, we have presented a regularization objective for inverse reinforcement learning to recover reward functions which are robust to spurious correlations present in expert datasets which feature diverse demonstrations. The robustness manifests itself as improved policy performance in a transfer setting in both the maximum entropy IRL case based on feature expectation matching as well as the adversarial setting. **Limitations and future work.** The hyperparameter $\lambda_I$ is strongly dependent on the data and environment – finding a automatic tuning procedure would overcome the main limitation in terms of the applicability of the method. Currently, the proposed method relies on a linear formulation of the causal invariance penalty. Successor methods of (Arjovsky et al., 2019) which introduce a nonlinear formulation (Lu et al., 2022) or include a larger class of distribution shifts (Rothenhäusler et al., 2021) could be considered in order to improve the overly conservative nature (Ahuja et al., 2021) of causally invariant features.

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

## A Proposition proofs

**Proposition 1.** *Let the likelihood $p(\xi)$ belong to a natural exponential family with parameter $\psi$, sufficient statistics $\varphi(x)$ and the (Lebesgue) base measure $p_0$. Let $\mathcal{D}_E^e$ be the dataset corresponding to interventional setting $e$. Then, for all $e \in \mathcal{E}_{tr}$, the causal invariance penalty for the maximum likelihood loss is the norm of the sufficient statistics expectation difference:*

$$\mathbb{D}(\psi, \varphi; e) = ||\nabla_{\psi|\psi=1.0}\mathcal{L}^e(\psi, \varphi)||^2 = ||\mathbb{E}_{\mathcal{D}_E^e}[\varphi(\xi)] - \mathbb{E}_{p(\xi|\psi)}[\varphi(\xi))]||^2 \tag{15}$$

*Proof.* The result directly follows from the definition of the primal problem.

$$\begin{aligned}
\nabla_{\psi|\psi=1.0}\mathcal{L}^e(\psi, \varphi) &= \nabla_\psi \left( \mathbb{E}_{\xi \in \mathcal{D}_e} \left[ \log \left( \frac{1}{Z_{\psi,\varphi}} \exp(\psi^T \varphi(\xi)) \right) \right] \right) \\
&= \mathbb{E}_{\xi \in \mathcal{D}_e} \left[ \nabla_\psi (\psi^T \varphi(\xi) - \log Z_{\psi,\varphi}) \right] \\
&= \mathbb{E}_{\mathcal{D}_E^e}[\varphi(\xi)] - \nabla_\psi \log Z_{\psi,\varphi} \\
&= \mathbb{E}_{\mathcal{D}_E^e}[\varphi(\xi)] - \mathbb{E}_{p(\xi|\psi)}[\varphi(\xi))]
\end{aligned}$$

where we use the moment generating property of the log partition function $\nabla_\psi \log Z_{\psi,\varphi} = \mathbb{E}_{p(\xi|\psi)}[\varphi(\xi)]$. $\square$

**Proposition 2.** *The gradient of the primal exponential family maximum-likelihood problem in Equation (3) w.r.t. the natural parameter $\psi$ is equivalent to the gradient of the dual in Equation (5) w.r.t the parameter $\psi$ when the density ratio $\frac{q}{p}$ is unity.*

$$||\nabla_\psi \mathcal{L}_{dual}(\psi, \varphi, q, e)||_2 = ||\min_q \mathbb{E}_{\xi \sim \mathcal{D}_E}[\varphi(\xi)] - \mathbb{E}_{\xi \sim p(\xi|\psi,\varphi)} \left[ \frac{q(\xi)}{p(\xi|\psi,\varphi)} \varphi(\xi) \right] ||_2$$

*Proof.* We begin by computing the gradient of

$$\min_q \left[ \mathbb{E}_{\xi \sim \mathcal{D}_E^e}[g_{\psi,\varphi}(\xi)] - \mathbb{E}_{\xi \sim q}[g_{\psi,\varphi}(\xi)] + D_{KL}(q||p_0) \right]$$

w.r.t. to the parameters $\psi$ for setting $e \in \mathcal{E}_{tr}$:

$$\begin{aligned}
\nabla_\psi \mathcal{L}_{dual}(\psi, \varphi, q, e) &= \nabla_\psi (\min_q \left[ \mathbb{E}_{\xi \sim \mathcal{D}_E^e}[g_{\psi,\varphi}(\xi)] - \mathbb{E}_{\xi \sim q}[g_{\psi,\varphi}(\xi)] + D_{KL}(q||p_0) \right]) \\
&\overset{(1)}{=} \nabla_\psi (\min_q \left[ \mathbb{E}_{\xi \sim \mathcal{D}_E}[g_{\psi,\varphi}(\xi)] - \mathbb{E}_{\xi \sim q}[g_{\psi,\varphi}(\xi)] + \mathcal{H}(q) \right]) \\
&\overset{(2)}{=} \min_q \left[ \nabla_\psi \left( \mathbb{E}_{\xi \sim \mathcal{D}_E}[g_{\psi,\varphi}(\xi)] - \mathbb{E}_{\xi \sim q}[g_{\psi,\varphi}(\xi)] + \mathcal{H}(q) \right) \right] \\
&\overset{(3)}{=} \min_q \left[ \mathbb{E}_{\xi \sim \mathcal{D}_E^e}[\nabla_\psi g_{\psi,\varphi}(\xi)] - \mathbb{E}_{\xi \sim q}[\nabla_\psi g_{\psi,\varphi}(\xi)] \right]
\end{aligned}$$

where we use: (1) the fact that we assume the base measure $p_0$ to be the Lebesgue measure (or count measure in the discrete case), i.e. $p_0(\xi) = 1$, (2) the envelope theorem and (3) the fact that $q$ does not directly depend on $\psi$ and thus, $\nabla_\psi \mathcal{H}(q) = 0$. We can rewrite the second expectation using the importance sampling trick:

$$\min_q \mathbb{E}_{x \sim \mathcal{D}_E}[\varphi(\xi)] - \mathbb{E}_{x \sim q}[\varphi(\xi)] = \min_q \mathbb{E}_{x \sim \mathcal{D}_E}[\varphi(\xi)] - \mathbb{E}_{x \sim p(\xi|\psi,\varphi)} \left[ \frac{q(\xi)}{p(\xi|\psi,\varphi)} \varphi(\xi) \right]$$

By definition of the distribution matching problem and strict concavity w.r.t. $q$, the optimum is attained when $q(\xi) = p(\xi|\psi, \varphi)$, i.e. when the importance sampling ratio is 1.

$\square$

# B    Additional results

In this section, we present additional experimental evidence to support the claims made in the main text.

## B.1    Additional gridworld results

Figure 5 depicts an additional experimental setting using algorithm 1. Here, we choose 5 sets of trajectories which solve a horizontal navigation problem illustrated in fig. 5a. We compare this to a same set of baselines as described in section 3.1 with the addition of a spectral norm (Yoshida & Miyato, 2017) regularizer which imposes the Lipschitz smoothness penalty on the reward representation. We motivate this choice by the fact that Lipschitz smoothness is a successful regularization techniques in the approximate setting. We can observe a similar pattern to the one reported in section 3.1, where the ERM MaxEnt baseline overfits to reward to the observed trajectories (fig. 5(b,c)). The Lipschitz regularization in fig. 5(d) provides a more succinct reward representation but fails to capture the horizontal reward gradient which is indicative of the shared intent of the experts to move right in the direction of the goal. The CI penalty with both regularization strengths (fig. 5(e,f)) recovers this aspect of the ground truth reward.

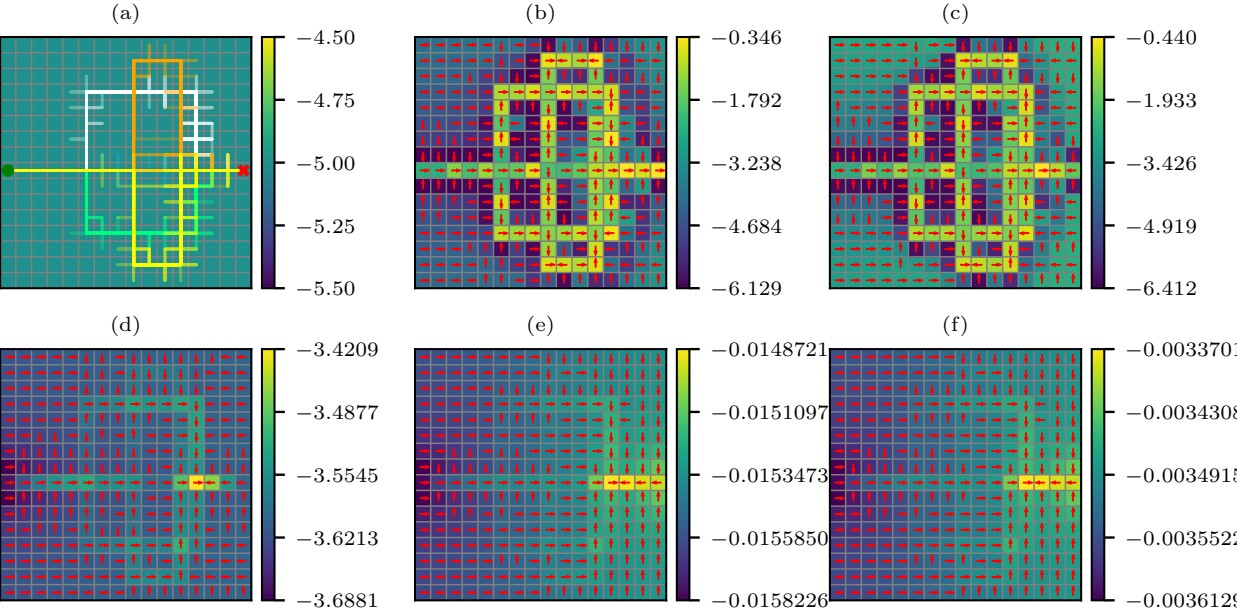

Figure 5: Feature matching reward recovery on a gridworld environment. (a) expert trajectory datasets: every color represents a modality containing 50 trajectories (b) MaxEnt IRL ERM baseline (c) MaxEnt IRL ERM baseline with L2 regularization coefficent $\lambda_{L2} = 1e-3$ (d) MaxEnt IRL baseline with spectral norm (Lipschitz) regularization (e) MaxEnt IRL with CI penalty, $\lambda_I = 0.1$, (f) MaxEnt IRL with CI penalty, $\lambda_I = 0.5$

## B.2    Adversarial training results

For reference, in fig. 6, we present the results of the adversarial training procedure used to recover the reward functions for the experiments in section 3.2. We can observe that when used to regularize the discriminator in an adversarial setting, the CI penalty does not produce a significant regularization effect as has been established in the transfer setting.

## B.3    Alternative discriminator training

Similar to the training dynamics reported in fig. 3 for the AIRL discriminator structure, we provide the results for the other two algorithms – GAIL in fig. 7 and MEIRL in fig. 8. We observe that for GAIL, the

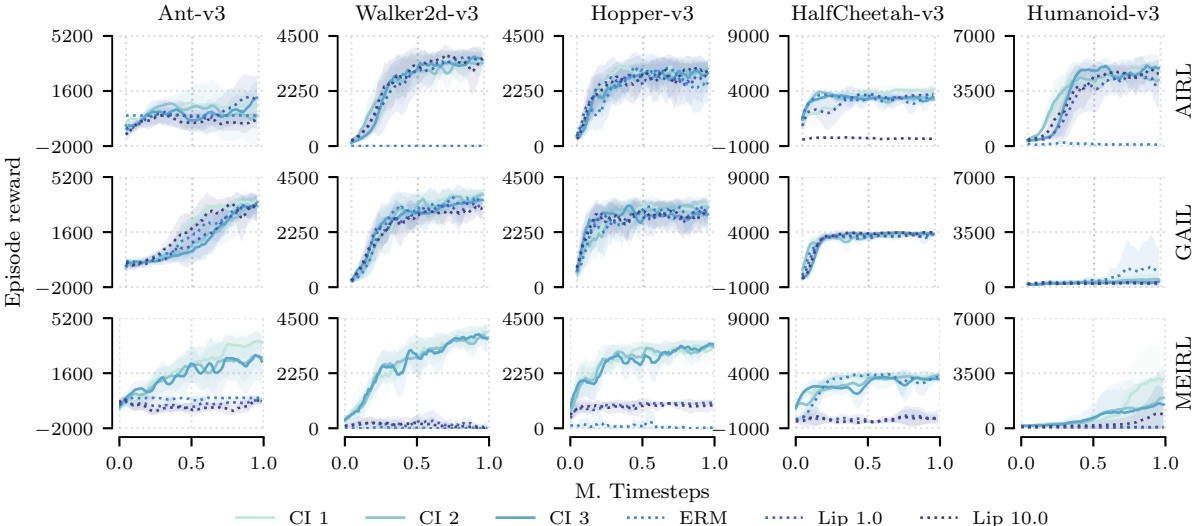

Figure 6: Comparison of SAC policy training performance w.r.t. ground truth reward when trained using the adversarial training procedures. Every row depicts a different type of algorithm used to train the policies.

regularization is beneficial in all settings except for HALFCHEETAH where it is outperformed by the Lipschitz regularized baseline. For the MEIRL setting, we also observe a significant improvement with the exception of the ANT environment, where all algorithm fail to achieve movement using the recovered reward.

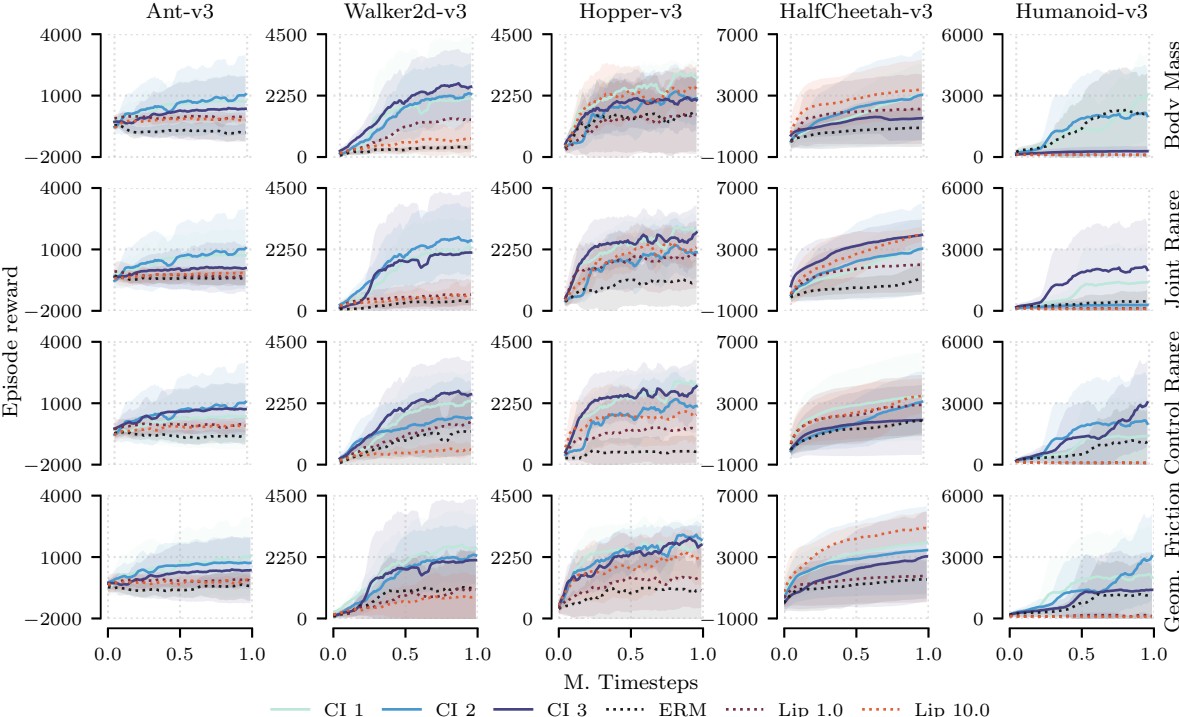

Figure 7: Comparison of SAC policy training performance w.r.t. ground truth reward when trained on recovered reward functions. Every row depicts a different type of dynamics perturbation for the five MuJoCo tasks as described in section 3.2. Here, GAIL is chosen as the baseline.

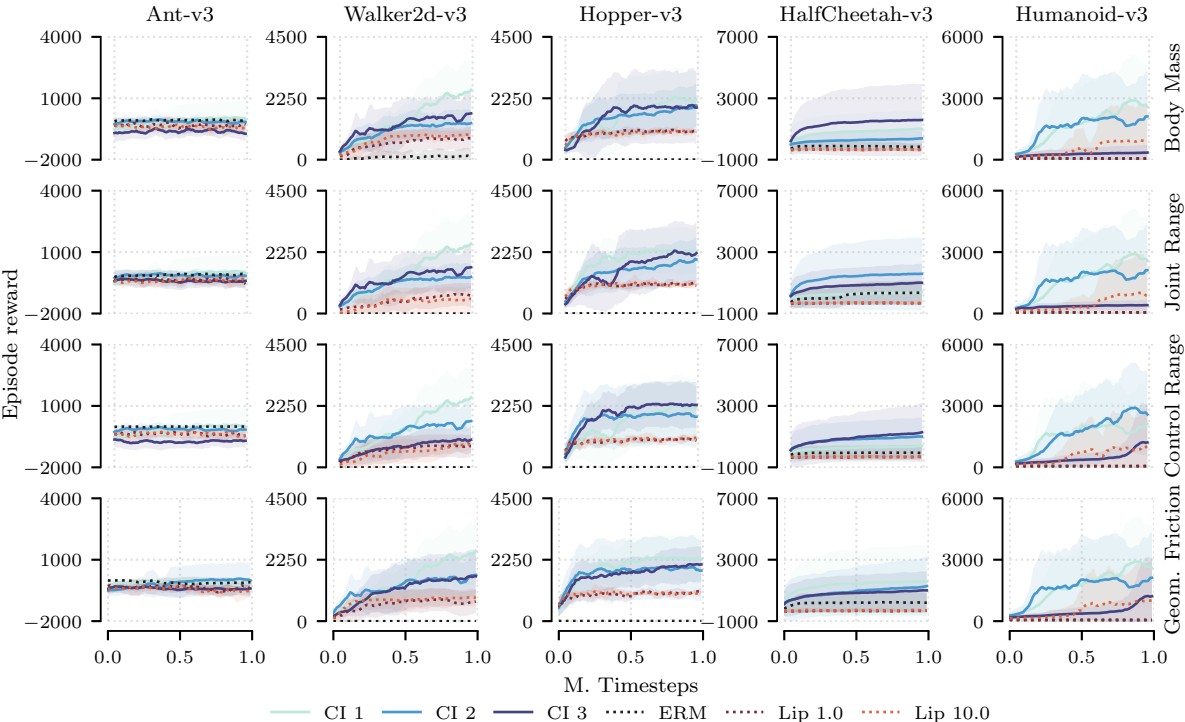

Figure 8: Comparison of SAC policy training performance w.r.t. ground truth reward when trained on recovered reward functions. Every row depicts a different type of dynamics perturbation for the five MuJoCo tasks as described in section 3.2. Here, MEIRL is chosen as the baseline.

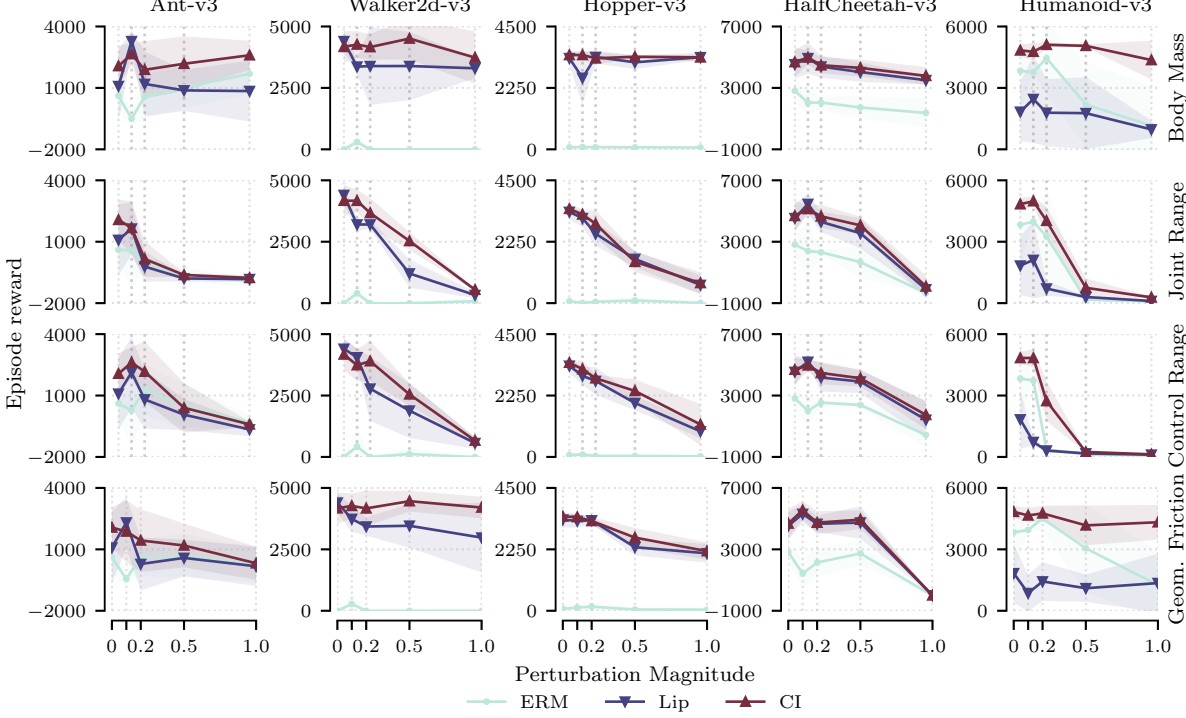

Figure 9: Comparison of SAC policy performance w.r.t. ground truth reward when trained on recovered reward functions as a function of the perturbation strength. Every row depicts a different type of dynamics perturbation for the five MuJoCo tasks as described in section 3.2. Here, AIRL is chosen as the baseline.

Table 2: Policy rollout results using ground truth reward for perturbed MuJoCo environments after being trained for 1M timesteps using the rewards recovered from the different discriminators in section 3.2. Here, the *joint range* parameter is perturbed with a noise magnitude of $\varepsilon = 0.2$. The results are averaged over 10 rollouts and obtained by training the model using five different random seeds.

| Environment | ANT-V3 | WALKER2D-V3 | HOPPER-V3 | HALFCHEETAH-V3 | HUMANOID-V3 |
|---|---|---|---|---|---|
| Expert | $3168.49_{\pm1715.68}$ | $3565.33_{\pm527.40}$ | $3119.54_{\pm524.36}$ | $4340.61_{\pm2020.14}$ | $4774.17_{\pm2063.52}$ |
| AIRL (ERM) | $-18.73_{\pm255.23}$ | $-3.50_{\pm1.66}$ | $48.82_{\pm76.24}$ | $2310.93_{\pm118.75}$ | $3261.17_{\pm2272.68}$ |
| AIRL (Lip) | $-213.89_{\pm738.75}$ | $3202.09_{\pm185.98}$ | $2544.28_{\pm445.86}$ | $4293.70_{\pm666.33}$ | $710.88_{\pm356.04}$ |
| AIRL (CI) | $155.62_{\pm875.01}$ | $3670.21_{\pm599.00}$ | $2906.77_{\pm490.33}$ | $4653.89_{\pm762.09}$ | $4022.43_{\pm671.37}$ |
| GAIL (ERM) | $-486.05_{\pm388.06}$ | $359.23_{\pm254.85}$ | $1047.76_{\pm871.98}$ | $1160.78_{\pm1134.26}$ | $508.86_{\pm601.99}$ |
| GAIL (Lip) | $-208.97_{\pm252.99}$ | $577.40_{\pm550.86}$ | $2339.87_{\pm465.97}$ | $4168.13_{\pm472.22}$ | $122.49_{\pm82.43}$ |
| GAIL (CI) | $1021.26_{\pm1845.56}$ | $3479.99_{\pm1242.39}$ | $2976.49_{\pm417.33}$ | $5581.43_{\pm1442.39}$ | $2170.96_{\pm2425.51}$ |
| MEIRL (ERM) | $-57.30_{\pm93.23}$ | $-3.69_{\pm0.18}$ | $3.17_{\pm0.52}$ | $347.55_{\pm790.54}$ | $54.70_{\pm1.92}$ |
| MEIRL (Lip) | $-554.25_{\pm82.05}$ | $622.45_{\pm464.25}$ | $1136.95_{\pm209.97}$ | $-297.84_{\pm71.23}$ | $1014.42_{\pm1915.24}$ |
| MEIRL (CI) | $-337.17_{\pm1310.57}$ | $2292.94_{\pm1521.43}$ | $2800.37_{\pm666.09}$ | $1650.50_{\pm1683.38}$ | $2935.90_{\pm2262.13}$ |

Table 3: Policy rollout results using ground truth reward for perturbed MuJoCo environments after being trained for 1M timesteps using the rewards recovered from the different discriminators in section 3.2. Here, the *actuator control range* parameter is perturbed with a noise magnitude of $\varepsilon = 0.2$. The results are averaged over 10 rollouts and obtained by training the model using five different random seeds.

| Environment | ANT-V3 | WALKER2D-V3 | HOPPER-V3 | HALFCHEETAH-V3 | HUMANOID-V3 |
|---|---|---|---|---|---|
| Expert | $3168.49_{\pm1715.68}$ | $3565.33_{\pm527.40}$ | $3119.54_{\pm524.36}$ | $4340.61_{\pm2020.14}$ | $4774.17_{\pm2063.52}$ |
| AIRL (ERM) | $1279.87_{\pm1281.66}$ | $-0.41_{\pm3.07}$ | $37.62_{\pm62.71}$ | $2536.90_{\pm212.27}$ | $357.19_{\pm135.66}$ |
| AIRL (Lip) | $809.60_{\pm1425.76}$ | $2779.73_{\pm1303.91}$ | $2784.28_{\pm510.50}$ | $4175.49_{\pm918.92}$ | $312.24_{\pm90.05}$ |
| AIRL (CI) | $2166.58_{\pm1471.70}$ | $3897.58_{\pm831.24}$ | $2884.34_{\pm130.60}$ | $4470.17_{\pm731.81}$ | $2730.60_{\pm982.13}$ |
| GAIL (ERM) | $-641.93_{\pm284.65}$ | $1180.57_{\pm1413.21}$ | $491.27_{\pm565.63}$ | $1862.67_{\pm1026.74}$ | $1219.94_{\pm1784.26}$ |
| GAIL (Lip) | $-92.74_{\pm363.44}$ | $1672.06_{\pm1263.95}$ | $2028.07_{\pm1004.96}$ | $3638.51_{\pm1164.42}$ | $93.01_{\pm24.01}$ |
| GAIL (CI) | $2486.00_{\pm2078.11}$ | $2660.74_{\pm866.36}$ | $2985.44_{\pm280.70}$ | $3979.90_{\pm2494.31}$ | $2986.70_{\pm2389.78}$ |
| MEIRL (ERM) | $-10.63_{\pm3.35}$ | $-3.83_{\pm0.45}$ | $3.17_{\pm0.27}$ | $-36.66_{\pm489.57}$ | $58.40_{\pm0.15}$ |
| MEIRL (Lip) | $-411.65_{\pm244.20}$ | $832.80_{\pm311.20}$ | $1073.22_{\pm138.00}$ | $-261.74_{\pm164.78}$ | $1064.41_{\pm2011.91}$ |
| MEIRL (CI) | $133.50_{\pm969.39}$ | $2286.80_{\pm1040.59}$ | $2551.59_{\pm1131.76}$ | $3303.82_{\pm2332.89}$ | $3058.78_{\pm2286.41}$ |

## B.4 Tables of results

In addition to the results reported on the body mass parameter of the MuJoCo simulation, we provide the results tables for other three perturbation parameters: *joint range* in table 2, *actuator control range* in table 3 and *geometry friction* in table 4. We observe a similar effect in terms of the reward generalization properties as described in the main text.

Table 4: Policy rollout results using ground truth reward for perturbed MuJoCo environments after being trained for 1M timesteps using the rewards recovered from the different discriminators in section 3.2. Here, the *geometry friction* parameter is perturbed with a noise magnitude of $\varepsilon = 0.2$. The results are averaged over 10 rollouts and obtained by training the model using five different random seeds.

| Environment | ANT-V3 | WALKER2D-V3 | HOPPER-V3 | HALFCHEETAH-V3 | HUMANOID-V3 |
|---|---|---|---|---|---|
| Expert | $3168.49_{\pm1715.68}$ | $3565.33_{\pm527.40}$ | $3119.54_{\pm524.36}$ | $4340.61_{\pm2020.14}$ | $4774.17_{\pm2063.52}$ |
| AIRL (ERM) | $603.00_{\pm909.86}$ | $-3.87_{\pm0.44}$ | $149.17_{\pm151.96}$ | $2141.85_{\pm942.19}$ | $4507.23_{\pm659.04}$ |
| AIRL (Lip) | $283.73_{\pm1294.15}$ | $3429.17_{\pm372.29}$ | $3311.25_{\pm128.82}$ | $4659.44_{\pm533.32}$ | $1432.57_{\pm952.87}$ |
| AIRL (CI) | $1434.01_{\pm1530.65}$ | $4167.15_{\pm721.26}$ | $3288.86_{\pm149.76}$ | $4737.72_{\pm749.52}$ | $4756.93_{\pm368.15}$ |
| GAIL (ERM) | $-421.27_{\pm752.40}$ | $910.12_{\pm951.80}$ | $939.05_{\pm959.38}$ | $1563.17_{\pm1245.51}$ | $871.61_{\pm1002.21}$ |
| GAIL (Lip) | $-172.69_{\pm196.92}$ | $1065.02_{\pm1672.12}$ | $2541.08_{\pm906.67}$ | $4795.59_{\pm1018.65}$ | $89.51_{\pm16.87}$ |
| GAIL (CI) | $1148.32_{\pm1938.45}$ | $2395.43_{\pm1282.70}$ | $3068.00_{\pm459.50}$ | $4037.08_{\pm983.32}$ | $3385.58_{\pm2279.28}$ |
| MEIRL (ERM) | $-103.10_{\pm188.90}$ | $-3.43_{\pm0.15}$ | $3.36_{\pm0.19}$ | $191.23_{\pm630.81}$ | $56.41_{\pm2.24}$ |
| MEIRL (Lip) | $-252.29_{\pm184.32}$ | $865.84_{\pm308.25}$ | $1128.52_{\pm125.06}$ | $-284.92_{\pm204.61}$ | $991.70_{\pm1871.72}$ |
| MEIRL (CI) | $-191.76_{\pm912.46}$ | $2546.37_{\pm1073.22}$ | $2425.38_{\pm1070.78}$ | $1724.44_{\pm2057.07}$ | $2155.55_{\pm2281.06}$ |

### B.5   Lunar Lander environment

In addition to the robotic locomotion setting, we have performed a set of reward learning experiments in the context of the LUNARLANDER environment (Towers et al., 2023). Similarly to the robotic locomotion experiments described in section 3.2, 10 demonstrations were gathered using a pre-trained SAC baseline and a structured noise approach was used to achieve diversity of demonstrations. The reward was trained using the AIRL baseline algorithm and the same set of regularization coefficients $\lambda_I \in \{0.0, 0.1, 1.0, 10.0, 100.0\}$ and $\lambda_L \in \{0.0, 1.0, 10.0\}$ as in section 3.2. Subsequently, the reward was used to train an SAC policy under dynamics perturbations on three different parameters: gravity, wind power and turbulence power with noise magnitudes $\eta = 1.0$ for the gravity and wind power parameters and $\eta = 0.5$ for the turbulence power parameter.

Figure 10 depicts the resulting policy training performance. We can observe that using the CI regularization with regularization coefficient $\lambda = 100.0$ allows us to train a policy which considerably improves upon both the unregularized baseline (ERM) and the Lipschitz smoothness regularized baselines (Lip) both in terms of faster convergence and asymptotic performance measured using the ground truth reward.

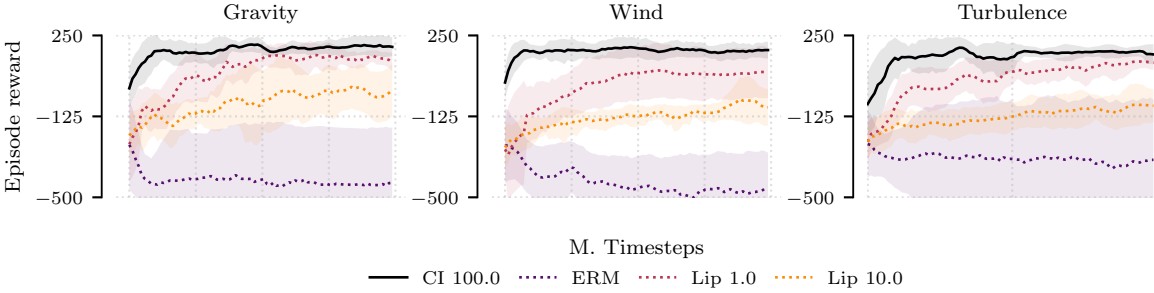

Figure 10: Comparison of SAC policy training performance w.r.t. ground truth reward when trained using the recovered reward functions. Here, AIRL is chosen as the baseline reward learning algorithm. Every column depicts a different type of dynamics perturbation used during training.

## C   Model architecture and training details

**Gridworld.**   For the gridworld experiments, we use a DeepMaxEnt (Wulfmeier et al., 2015) formulation of the IRL problem. The state features are parametrized by a 2-layer MLP with a 1-dimensional hidden layer. We use RMSProp as an optimizer with a learning rate of $1e-3$.

**MuJoCo and LunarLander.**   For the adversarial learning experiments, we use an actor and critic network with two hidden layers of size 256 and a discriminator network with two hidden layers of size 128. We use Adam (Kingma & Ba, 2014) as the optimizer with an initial learning rate of $\eta = 3e-4$. We use the gradient penalty strategy described in (Gulrajani et al., 2017) to impose the Lipschitz smoothness constraint on the discriminator networks in section 3.2. We perform one update of the discriminator network for every update of the actor-critic networks. We use an extension of the cleanRL (Huang et al., 2022) library for all the experiments. to train the policy networks both using the ground truth reward as well as during adversarial training and using the recovered reward for the transfer experiments.

### C.1   Regularization strength

The causal invariance (CI) penalty term used in eq. (14) and eq. (16) bears resemblance to the Lipschitz smoothness gradient penalty (Gulrajani et al., 2017) used as the main regularized baseline througout the experiments presented in this work. The magnitude of both penalty types is controlled by the respective regularization coefficients $\lambda_L$ and $\lambda_I$. The regularization strength of the CI gradient penalty controls the deviation of the learned solution from the optimum on the respective training setting in the convex case (Arjovsky et al., 2019). In contrast to penalizing the smoothness of the function with respect to the input, the

tuning of the regularization is largely dependent on the diversity of training settings. Identifying a data-driven strategy for regularization tuning is a challenge we defer to future work.

## D  Derivation of the penalty term for feature matching IRL

To apply the proposed regularization to the feature matching problem described in algorithm 1, we need to derive an explicit gradient term. We will do so here. The gradient of the per-environment log-likelihood loss $\mathcal{L}_{MLE}(\psi, \varphi, e) = \sum_{\xi \in \mathcal{D}_e} \log p(\xi|\psi, \varphi)$ w.r.t. to $\psi$ is computed as follows:

$$\mathcal{L}_{MLE} = \sum_{\xi \in \mathcal{D}_e} \log p(\xi|\psi, \varphi) = \sum_{\xi \in \mathcal{D}_e} \log(\exp(\psi^T \varphi(\xi))) - \log Z_{\psi, \varphi} = \sum_{\xi \in \mathcal{D}_e} \psi^T \varphi(\xi) - \log Z_{\psi, \varphi}$$

Differentiating w.r.t. $\psi$ yields:

$$\frac{\partial \mathcal{L}_{MLE}}{\partial \psi} = \mathbb{E}_{\mathcal{D}_e}[\varphi(\xi)] - \frac{1}{Z_{\psi, \varphi}} \int \exp(\psi^T \varphi(\xi)) \varphi(\xi) d\xi$$

$$= \mathbb{E}_{\mathcal{D}_E}[\varphi(\xi)] - \mathbb{E}_{p(\xi|\psi)}[\varphi(\xi)]$$

The gradient penalty term from Equation (15) with respect to the features $\varphi$ is derived as follows:

$$\nabla_\varphi \left\| \nabla_{\psi|\psi=1.0} \mathcal{L}^e \left( r\left(\psi, \varphi\right)\right) \right\|^2 = \frac{\partial \left\| \frac{\partial \mathcal{L}^e(r(\psi, \varphi) \cdot)}{\partial \psi} \big|_{\psi=1.0} \right\|^2}{\partial \varphi}$$

We employ the chain rule:

$$\frac{\partial \mathcal{L}^e \left( r\left(\psi, \varphi\right)\right)}{\partial \psi} = \frac{\partial \mathcal{L}^e \left( r\left(\psi, \varphi\right)\right)}{\partial r} \cdot \frac{\partial \left( r\left(\psi, \varphi\right)\right)}{\partial \psi} = \frac{\partial \mathcal{L}^e \left( r\left(\psi, \varphi\right)\right)}{\partial r} \cdot \varphi$$

where the last equality holds because we assume a linear reward with respect to the features $\varphi$: $r\left(\psi, \varphi\right) = \psi^T \varphi$. In section 2.1 we showed that :

$$\frac{\partial \mathcal{L}^e \left( r\left(\psi, \varphi\right)\right)}{\partial r} = \mathbb{E}_{\mathcal{D}_E}[\varphi(\xi)] - \mathbb{E}_{p(\xi|\psi)}[\varphi(\xi)]$$

where $\mathbb{E}_{\mathcal{D}_E}[\varphi(\xi)]$ are the trajectory feature statistics of the expert and $\mathbb{E}_\pi[\varphi(\xi)]$ are the trajectory feature statistics of the imitation policy. For the sake of simplicity we define: $C := \mathbb{E}_{\mathcal{D}_E}[\varphi(\xi)] - \mathbb{E}_{p(\xi|\psi)}[\varphi(\xi)]$ which is independent of $\varphi$: Then:

$$\frac{\partial \mathcal{L}^e \left( r\left(\psi, \varphi\right)\right)}{\partial \psi} = C\varphi$$

We obtain the CI penalty for the feature matching maximum entropy IRL case as the following term:

$$\nabla_\varphi \left\| \nabla_{\psi|\psi=1.0} \mathcal{L}^e \left( r\left(\psi, \varphi\right)\right) \right\|^2 = \frac{\partial \left\| \frac{\partial \mathcal{L}^e(r(\psi, \varphi) \cdot)}{\partial \psi} \big|_{\psi|\psi=1.0} \right\|^2}{\partial \varphi} = \frac{\partial \left\| C\varphi \right\|^2}{\partial \varphi}$$

$$= \frac{\partial \left[ C\varphi \right]^T \left[ C\varphi \right]}{\partial \varphi}$$

$$= C^T C \frac{\partial \varphi^T \varphi}{\partial \varphi} = 2 \left\| C \right\|^2 \varphi = 2||\rho_E - \mathbb{E}\left[\rho_\psi\right]||^2 \varphi$$

