# OpenReview forum: "Learning Causally Invariant Reward Functions from Diverse Demonstrations"
_TMLR — Rejected by TMLR_

### Review · Reviewer_5zTi · 2024-06-17

**Summary Of Contributions:**

This paper tackles a challenge in inverse reinforcement learning: when performing reward learning, the expert demonstrations can have spurious correlations that lead to inferior performance. To solve this difficulty, the authors propose to leverage the causal invariance properties in expert demonstrations, which device a proper regularization in the reward learning objective. Through numerical results, the paper shows the performance of the proposed method.

**Audience:**

Yes

**Claims And Evidence:**

No

**Requested Changes:**

I would recommend a rewriting of Section 2 and move the related work section forward to discuss the contributions.

There are typos and grammatical issues in places across the paper. I would suggest the authors to make a thorough pass of the manuscript.

**Strengths And Weaknesses:**

Strengths: The challenges highlighted in the introduction section is of interest. Solving these challenges will definitely have an impact on improving inverse reinforcement learning.

There are plenty of numerical results and the performance of the proposed method seems to suggest some promises.

Weakness: I am a none expert on causal inference and inverse reinforcement learning. I have spent quite a few back-and-forth in Section 2 to follow the logic and derivation. However, there are still quite many gaps for me to fully understand it. Firstly, there are some undefined notations. For example, Eq. (5) involves an undefined $q$ in the first equality and the second equality is not explained very clearly. Another example is that $\phi$ and $\varphi$ are used interchangeably. While my biggest complaint is about the novelty in the method. It makes quite dirty to judge what are the unique developments in Algorithm 1 and 2, given my limited background.

There are actually several challenges highlighted in the introduction section. The paper proceeds with "to circumvent this issue", referring to the spurious correlation challenge. I am wondering whether other challenges are solved, such as sample complexity issue.

---

> ### Author Response · Authors · 2024-07-22
>
> _We thank the reviewer for the comments and constructive suggestions. We appreciate the fact that our work is deemed relevant and that the promising performance of our method is of note._
>
> - There are actually several challenges highlighted in the introduction section. The paper proceeds with "to circumvent this issue", referring to the spurious correlation challenge. I am wondering whether other challenges are solved, such as sample complexity issue.
>
> _While we do mention several challenges common in inverse RL, we primarily tackle the reward generalization problem. We do not specifically address the sample complexity issue, however we do use an off-policy algorithm to train the policy - doing so results in improved sample efficiency compared to on-policy versions of GAIL and AIRL which use TRPO._
>
> _We have made a pass over the manuscript and streamlined section 2._

---

### Review · Reviewer_htZy · 2024-07-07

**Summary Of Contributions:**

This paper incorporates the principle of Causal Invariance (CI) as a regularization of the IRL objective, in order to recover a reward that is robust to distributional shifts of the dynamics.

For that, they augment the entropy-regularized objective (Eq.6) with a penalty on the gradient of an 'expected loss' as seen in Eq.16

Just like the entropy-regularized objective, this one can be directly optimized in the discret MDP case (Algo1), or interpreted as a generative adversarial learning problem (like GAIL or AIRL) for the approximation setting (Algo2).

Adding that augmentation for CI regularization can thus be easily branched on top of existing approaches (GAIL + CI, AIRL + CI, MEIRL + CI).

**Audience:**

Yes

**Broader Impact Concerns:**

Does not apply.

**Claims And Evidence:**

Yes

**Requested Changes:**

*Regarding IRL:*

When describing MaxEntIRL, wouldn't be much easier for an unfamiliar reader to start by saying
" we learn the reward as a scalar product R_\phi=<\phi, \psi>, where \phi represents state features, that can be learned or are sometimes directly specified " ?  (instead of the first sentence "the first-order feature statistics of the trajectory ... is matched with the statistics of ..." which is very hard to interpret)

In Eq.4, what exactly is p_0 ? A kind of a-priori knowledge on the distribution of experts trajectories? and since in the end we are only interested in the entropy-regularization case (p_0 uniform), why not remove it to make things more simple (with entropies instead of KLs)?

In Eq.5, what is the dependency of the KL term in the optimized variable \phi? more specifically, what is q ??

I am not sure about the usefulness of the subsection "connection to f-divergences".

*Regarding CI*

If I understand correctly, the SCM as described is a acyclic graphical model where p(x | pa(x) ) is determined by functions f(x, pa(x), \epsilon) where \epsilon is a random variable ~ P(\epsilon). If this is just that, the description could be quite simpler.

In 'transitions SCM', what does e\inI(\epsilon_tr) means? is 'e' the index of an expert randomly sampled (following a specific perturbed dynamic)?
If so, I understand that we don't want the inferred optimality variable O to have a causal dependency in E, since O should be independent on the choice of the expert. But then, I don't understand Fig1 at all. In all cases, there is a causality path between E and O, since O depends on the action, that depends on C, that depends on E.

Also in Fig1, why are the actions unobserved ? in IRL, actions being observed is the very first assumption.
What does "stable conditional" (in blue) mean?
I don't get at all the meaning of the colors.
What is a blackdoor path in that case? why is such a path a problem, and why are the red arrows defining such a path?
Why X^e and X^ne ? What does "ne" represent?

In Eq.10, and I think this is the very crucial missing explanation in the paper: What is the "expected loss" ? Is it the objective function without the regularization term?
Without understanding what is L^e (sometimes also noted L_e), how can anyone understand why adding a penalty on its gradient induces a regularization based on CI?

In 2.3 --> I am not sure to agree with the assumption that having experts that are diverse because they follow differently perturbed rewards leads to learning a reward that is generalizable to perturbed dynamics.

*Regarding experiments*
I would like to understand what caused that discrepancy in the learned reward function by the baselines in Fig2 (b) and (c). Given the expert trajectory in (a), I see no reasons the diagonal path would have an absorbing state in the middle.

*Details:*

In f-divergency section, just after "R_d=-D_f, "The allows" --> "This allows"
In 2.2 second paragraph, "is considered spurious it is ..." --> "is considered spurious IF if is ..."
In Algo1 [\phi(\zeta) ) --> [\phi(\zeta) ]

**Strengths And Weaknesses:**

*Strength:*

The idea is novel and makes sense.
Since it results in a few modifications of the objective of existing approaches, it is easy to implement and yet shows nice improvements.
The experimental results are impressive, and the experiments are well described.

*Weaknesses:*

The theoretical part, introducing the concept of IRL and especially of CI, needs a lot of clarifications.
There are too many variables or notations that are introduced without being explained.
As I already know about IRL and that I have read most of the cited papers, I could follow the IRL part.
However, being unfamiliar with CI, I was completely lost, starting at 2.2.
I develop that critic in the Requested Changes.

---

> ### Author Response · Authors · 2024-07-22
> **Reply to Reviewer htZy (1)**
>
> We would like to thank the reviewer for the detailed comments and address the requested changes below.
>
> Regarding IRL:
>
> - When describing MaxEntIRL, wouldn't be much easier for an unfamiliar reader to start by saying " we learn the reward as a scalar product R_\phi=<\phi, \psi>, where \phi represents state features, that can be learned or are sometimes directly specified " ? (instead of the first sentence "the first-order feature statistics of the trajectory ... is matched with the statistics of ..." which is very hard to interpret)
>
> _We appreciate the suggestion and have adapted the description in the revision accordingly._
>
> - In Eq.4, what exactly is p_0 ? A kind of a-priori knowledge on the distribution of experts trajectories? and since in the end we are only interested in the entropy-regularization case (p_0 uniform), why not remove it to make things more simple (with entropies instead of KLs)?
>
> _In an exponential family of distributions, $p_0$ describes the base measure of the distribution. In the context of IRL, it can be used to model an a-priori knowledge of the expert trajectories. We have streamlined the exposition according to the suggestion._
>
> - In Eq.5, what is the dependency of the KL term in the optimized variable \phi? more specifically, what is q ??
>
> _$q$ is the trajectory distribution induced by the imitation policy. It is not dependent on the optimized variable $\varphi$. In our case, we assume a uniform base measure. Hence, as the reviewer noted previously, we only consider the entropy of the sampling distribution._
>
> - I am not sure about the usefulness of the subsection "connection to f-divergences".
>
> _The main purpose of this subsection was to introduce a numerical solution method to optimize f-divergences using adversarial optimization and to motivate the regularization of the discriminator loss in the dual case. We have shortened it somewhat in the revision._
>
> - If I understand correctly, the SCM as described is a acyclic graphical model where p(x | pa(x) ) is determined by functions f(x, pa(x), \epsilon) where \epsilon is a random variable ~ P(\epsilon). If this is just that, the description could be quite simpler.
>
> _This is, in essence, correct. We had included the standard SCM definition as common in causal inference literature for purpose of completeness._
>
> - In 'transitions SCM', what does e\inI(\epsilon_tr) means? is 'e' the index of an expert randomly sampled (following a specific perturbed dynamic)? If so, I understand that we don't want the inferred optimality variable O to have a causal dependency in E, since O should be independent on the choice of the expert. But then, I don't understand Fig1 at all. In all cases, there is a causality path between E and O, since O depends on the action, that depends on C, that depends on E.
>
> _It is correct that we want to eliminate the dependence of $O_t$ on $E$. In order to do so, we want to find associations between features of the transition tuple $\varphi(s,a,s’)$ and the optimality label $O_t$, which are stable across the training settings / independent of $E$.
> The three subfigures show examples of spurious correlations which can be eliminated using the causal invariance principle._
>
> - Why X^e and X^ne ? What does "ne" represent?
>
> _In the most general case, one can assume an arbitrary (e.g. neural network feature) transformation of the transition tuple into causal features $x^{(c)}$ and non-causal features $x^(nc)$ w.r.t. the label $O_t$. For instance, a discriminator function typically uses such a feature transformation to make the optimality prediction.
> This scenario is depicted in Fig. 1b. In this case, conditioning on the non-causal feature variable $x^(nc)$ corresponds to a collider structure and creates another type of spurious association path in the graphical model._
>
> - What does "stable conditional" (in blue) mean?
>
> _In the general case, $O_t \not\perp E$. However, conditioning on $s_t$ and $a_t$ shields the optimality label $O_t$ from the influence of the environment, inducing the following conditional independence relationship:  $P(O_t \perp E | s_t, a_t)$. This is what we refer to as the stable conditional as it should be stable across training settings / expert demonstrations._
>
> - Also in Fig1, why are the actions unobserved ? in IRL, actions being observed is the very first assumption.
>
> _While we do not explicitly study this setting, unobserved actions correspond to the learning from observations modality, where only expert states are available. In practice, this is a common enough modification of the adversarial algorithms that we have decided to include it as an example in the context of the SCM. Furthermore, some formulations of AIRL use a state-only definition of the reward to ensure reward recovery ([1, Thm C.1])._
>
> _[1] Justin Fu, Katie Luo, and Sergey Levine. Learning robust rewards with adversarial inverse reinforcement learning._

---

> ### Author Response · Authors · 2024-07-22
>
> - What is a blackdoor path in that case? why is such a path a problem, and why are the red arrows defining such a path?
>
> _Due to the unobserved confounder variable $C$, not observing the actions breaks the conditional independence relationship and introduces a non-causal association path in the graphical model which is what we refer to as a backdoor path._
>
> - I don't get at all the meaning of the colors.
>
> _The colors are meant to indicate the two different types of spurious association paths: orange for collider conditioning and red for backdoor path._
>
> - In Eq.10, and I think this is the very crucial missing explanation in the paper: What is the "expected loss" ? Is it the objective function without the regularization term? Without understanding what is L^e (sometimes also noted L_e), how can anyone understand why adding a penalty on its gradient induces a regularization based on CI?
>
> _Eq. 10 describes the generic IRM formulation which is applicable to arbitrary loss functions $L^e$. In our case, this corresponds to the respective IRL objective function: log-likelihood in the primal case or classification loss in the dual case._
>
> _We have corrected this omission in the manuscript._
>
> - In 2.3 --> I am not sure to agree with the assumption that having experts that are diverse because they follow differently perturbed rewards leads to learning a reward that is generalizable to perturbed dynamics.
>
> _Our main intuition is that deriving a more succinct representation of the reward should enable an easier acquisition of behaviours.
> For instance - different styles of gait in locomotion all share a common reward feature, namely that of moving forward.
> Identifying this reward as opposed to absorbing different sources of variance from diverse expert demonstrations allows the reward to be more readily used in perturbed environments._
>
> Regarding Experiments
>
> - I would like to understand what caused that discrepancy in the learned reward function by the baselines in Fig2 (b) and (c). Given the expert trajectory in (a), I see no reasons the diagonal path would have an absorbing state in the middle.
>
> _We have performed additional experiments and conclude that the reason for the poor reward learning of the diagonal is due to the selection bias introduced as part of the experimental setup.
> For a multimodal setting with equal number of trajectories, we obtain more consistent results, which can be seen in the supplementary material. We have also re-run the selection bias experiment with a larger network which would provide a higher capacity to learn the middle modality. Despite the increased capacity the baselines still exhibit an absorbing state in the middle path. We have also included this experiment as part of the supplementary material._

---

### Review · Reviewer_9vUR · 2024-07-08

**Summary Of Contributions:**

This work addresses the problem of inverse reinforcement learning in the presence of demonstrations from a set of experts whose policies may have variability across expert policies. The authors draw a connection between the variability across expert demonstrations with invariance. Specifically, under the assumption that the reward function is stable the authors are able to invoke the intuition and regularization approach from the causal invariance literature. The authors then, in turn, propose using regularization to focus on portions of the conditional expectation that is invariant across experts which removes spurious correlations. The authors adapt existing causal invariance regularization approaches to the IRL setting and provide a variational formulation to avoid the computational burden of computing partition functions.

**Audience:**

Yes

**Claims And Evidence:**

Yes

**Requested Changes:**

It would be helpful if the authors could address weakness (3). If time allows it would also be useful to have a slightly expanded set of empirical evaluations, given the largely empirical nature of the paper.

**Strengths And Weaknesses:**

Strengths:
1. This work provides a very natural solution to one of the challenges of IRL in the presence of demonstrations from a collection of experts.
2. The paper is clearly written, the problem is well articulated, and the solution is presented in an easily accessible manner. Assumptions are laid plain, and the authors clearly delineate their points of contribution.
3. The proposed method shows strong empirical performance in the experiments.

Weaknesses:
1. While not a "weakness" in the strict sense of the word, the paper does not contain much theoretical characterizations of the proposed approach.
2. Given (1) it would be nice if there were demonstrations / experiments on a more extensive set of domains.
3. The regularization parameter in this setting is less easily interpreted than in standard regularization contexts, it would be useful if the authors provided some discussion.

---

> ### Author Response · Authors · 2024-07-22
>
> We thank the reviewer for the encouraging assessment of our work and would like to address the raised points below. We have addressed the requested weaknesses in the revised version of the manuscript by providing an additional evaluation setting in appendix B.5. We have performed experiments on the lunar lander environment under three different dynamics perturbation settings. The results corroborate our findings.
>
> The tuning of the regularization parameter is in a lot of ways similar to the tuning of the Lipschitz smoothness gradient penalty used as baseline throughout the presented work. However, as the causal invariance penalty measures the deviation from the optimal solution for a particular training setting $e\in\mathcal{E}_{tr}$, the regularization strength can be more data dependent due to the nature of the underlying interventions and the finite sample properties of the data in the respective training setting. As requested, we have provided a discussion of this aspect in appendix C.

---

### Decision · Action_Editor_nYLL · 2024-08-11

**Recommendation:** Reject

**Comment:**

The reviewers agree that the method is novel and interesting.
Two reviewers criticized the presentation for being difficult to read by explaining many relations that may not be relevant for the core contributions (e.g. the relations between adversarial imitation learning and f-divergence based distribution matching). However, reviewer 9vUR praised the clarity and noted that "the solution is presented in an easily accessible manner". At least to some extend, I agree with both perspectives: On the one hand, I think that it is comprehensible why and how the authors apply invariant risk minimization to IRL, but on the other hand I also think that some of the paragraphs (sometimes even among the more technical ones) feel a bit out of place. The authors tried to address this point by shortening the discussion on f-divergences, but I still think that the presentation could me more streamlined. For example, I do not see the point of explicitly mentioning the total variation distance in Eq. 8, given that CI-AIRL is based on AIRL which minimizes the reverse KL.

Overall, I request the following revisions:
* The authors should carefully reconsider how the method is introduced and which preliminaries need to be established. In my opinion, IRM and (deep) MaxEnt-IRL would already provide a sufficient basis to motivate and derive the proposed method. Additional preliminaries should be added, when they are necessary to convey particular insights.
* The paper should better introduce IRM and its application to (deep) MaxEnt-IRL
* It needs to be better justified why the policy update is not directly affected by the regularizer
* The submission also needs to discuss and empirically evaluate the effect of pooling the demonstrations vs. alternatively updating on a different expert set

I am afraid that these revisions will require another round of reviewing, and, therefore, I recommend to reject, but encourage the authors to resubmit after adequately addressing the aforementioned points.

**Audience:**

The proposed method is an interesting approach for learning more generalizable reward function with inverse reinforcement learning, which is an important open research questions. Hence, the method is of sufficient interest for the TMLR's audience.

**Claims And Evidence:**

The main claims are factually correct and the method is sound and principled in general. However, the paper lacks clarity, which also affects the narrative of treating the variations in the expert demonstrations as interventions, which could be formalized more explicitly. The discussion of the method itself also lacks clarity and could be improved by better focusing on those aspects that are more relevant to the proposed method, e.g. by justifying why the proposed regularizer does not directly affect the policy update.

**Resubmission Of Major Revision:**

The authors may consider submitting a major revision at a later time.